# No Free Lunch From Random Feature Ensembles:
# Scaling Laws and Near-Optimality Conditions

Benjamin S. Ruben [1]   William L. Tong [2 3 4]   Hamza Tahir Chaudhry [2 3]   Cengiz Pehlevan [2 3 4]

## Abstract

Given a fixed budget for total model size, one must choose between training a single large model or combining the predictions of multiple smaller models. We investigate this trade-off for ensembles of random-feature ridge regression models in both the overparameterized and underparameterized regimes. Using deterministic equivalent risk estimates, we prove that when a fixed number of parameters is distributed among $K$ independently trained models, the ridge-optimized test risk increases with $K$. Consequently, a single large model achieves optimal performance. We then ask when ensembles can achieve *near*-optimal performance. In the overparameterized regime, we show that, to leading order, the test error depends on ensemble size and model size only through the total feature count, so that overparameterized ensembles consistently achieve near-optimal performance. To understand underparameterized ensembles, we derive scaling laws for the test risk as a function of total parameter count when the ensemble size and parameters per ensemble member are jointly scaled according to a "growth exponent" $\ell$. While the optimal error scaling is always achieved by increasing model size with a fixed ensemble size, our analysis identifies conditions on the kernel and task eigenstructure under which near-optimal scaling laws can be obtained by joint scaling of ensemble size and model size.

## 1. Introduction

Ensembling methods are a well-established tool in machine learning for reducing the variance of learned predictors. While traditional ensemble approaches like random forests (Breiman, 2001) and XGBoost (Chen & Guestrin, 2016) combine many weak predictors, the advent of deep neural networks has shifted the state of the art toward training a single large predictor (LeCun et al., 2015). However, deep neural networks still suffer from various sources of variance, such as finite datasets and random initialization (Atanasov et al., 2022; Adlam & Pennington, 2020; Lin & Dobriban, 2021; Atanasov et al., 2024). As a result, deep ensembles—ensembles of deep neural networks—remain a popular method for variance reduction (Ganaie et al., 2022; Fort et al., 2020), and uncertainty estimation (Lakshminarayanan et al., 2017).

A critical consideration in practice is the computational cost associated with ensemble methods. While increasing the number of predictors in an ensemble improves its accuracy (provided each ensemble member is "competent" (Theisen et al., 2023)), each additional model incurs significant computational overhead. Supposing a fixed memory capacity for learned parameters, a more pragmatic comparison is between an ensemble of neural networks and a single large network with the same total parameter count. Indeed, recent studies have called into question the utility of deep ensembles relative to a single network of comparable total size (Abe et al., 2022; Vyas et al., 2023). However, because ensemble learning allows fully parallelized training and inference, it will likely remain an attractive option in practice. A robust understanding of when an ensemble of small neural networks might *approach* the performance of a single large neural network is therefore needed.

Originally introduced as a fast approximation to Kernel Ridge Regression, random-feature ridge regression (RFRR) (Rahimi & Recht, 2007) has emerged as a rich "toy model" for deep learning, capturing non-trivial effects of dataset size and network width (Canatar et al., 2021; Atanasov et al., 2023; Mei & Montanari, 2022). In particular, the generalization error of deep networks and RFRR can vary non-monotonically with dataset size and feature count due to over-fitting effects known as "double-descent" (D'Ascoli

[1] Biophysics PhD Program, Harvard University, Cambridge, MA 02138, USA [2] John A. Paulson School of Engineering and Applied Science, Harvard University, Cambridge, MA 02138, USA [3] Center for Brain Science, Harvard University, Cambridge, MA 02138, USA [4] Kempner Institute, Harvard University, Cambridge, MA 02138, USA. Correspondence to: Benjamin S. Ruben <ben-ruben@g.harvard.edu>.

*Proceedings of the 42$^{nd}$ International Conference on Machine Learning*, Vancouver, Canada. PMLR 267, 2025. Copyright 2025 by the author(s).

et al., 2020; Nakkiran, 2019; Nakkiran et al., 2019; Adlam & Pennington, 2020; Lin & Dobriban, 2021), or "multiple descent" (D'Ascoli et al., 2020; Mel & Ganguli, 2021; Meng et al., 2022). However, these overfitting effects can be mitigated with an optimally tuned ridge parameter (Nakkiran et al., 2020; Advani et al., 2020; Canatar et al., 2021; Simon et al., 2023). Specifically, Simon et al. showed that in RFRR, test risk decreases monotonically with model size and dataset size when the ridge parameter is optimally chosen.

In this present work, we study the tradeoff between the number of predictors and the size of each predictor in ensembles of RFRR models. We find that with a fixed budget on total model size (defined as the total number of random features), minimal error is achieved by a single large model, provided the ridge is optimally tuned (theorem 5.1). However, we identify multiple conditions under which ensembles of smaller models can achieve *near*-optimal performance. In the overparameterized regime, we show that, to leading order, error depends on the ensemble size $K$ and individual model size $N$ only through the total ensemble size $M = KN$ (eq. 24), so that overparameterized ensembles consistently achieve near-optimal performance. In the underparameterized regime, we derive scaling laws for the test risk as a function of total ensemble size (eq. 26). The scaling law depends on a newly introduced "growth exponent" $\ell$ which controls the relative growth rates of $K$ and $N$ with total model size $M$ (eq. 25). While optimal error scaling is always achieved by scaling model size $N$ with fixed $K$, we identify conditions on the kernel and task eigenstructure which permit near-optimal error scaling by joint scaling of ensemble size and model size.

## 2. Preliminaries

### 2.1. Random Features and the Kernel Eigenspectrum

In this section, we describe ensembled RFRR, as well as the spectral decomposition of the kernel on which our results rely. This framework is described in (Simon et al., 2023) for the single-predictor case, and is reviewed in more rigor in Appendix A. Here we will consider a supervised learning task, where the goal is to learn an estimator $\hat{f}(\boldsymbol{x})$ that maps input features $\boldsymbol{x} \in \mathbb{R}^D$ to a target value $y \in \mathbb{R}$, given a training set $\mathcal{D} = \{\boldsymbol{x}_p, y_p\}_{p=1}^P$. The training labels are assigned as $y_p = f_*(\boldsymbol{x}_p) + \epsilon_p$ for some ground truth function $f_*$, and where $\epsilon_p \sim \mathcal{N}(0, \sigma_\epsilon^2)$ is drawn i.i.d. for each sample.

**Random-Feature Ridge Regression (RFRR) Ensembles.** In a Random-Feature Ridge Regression (RFRR) Ensemble of size $K$, the data is first transformed into $K$ separate finite-dimensional feature-spaces. Linear regression is performed on each of these spaces independently, and the resulting predictions are averaged at test time. Throughout this paper,

we will use upper indices $k = 1, \ldots, K$ to represent the ensemble index. $K$ sets of random features $\boldsymbol{\psi}^k(\boldsymbol{x}) \in \mathbb{R}^N$ are constructed as $\left[\boldsymbol{\psi}^k(\boldsymbol{x})\right]_n = g(\boldsymbol{v}_n^k, \boldsymbol{x})$. Here, for each $n = 1, \ldots, N$ and $k = 1, \ldots, K$, $\boldsymbol{v}_n^k$ is a random parameter vector sampled independently as $\boldsymbol{v}_n^k \sim \mu_{\boldsymbol{v}}$ for measure $\mu_{\boldsymbol{v}}$ on $\mathbb{R}^C$. The function $g : \mathbb{R}^C \times \mathbb{R}^D \mapsto \mathbb{R}$ is a "featurization transformation," often taking the form $g(\boldsymbol{v}, \boldsymbol{x}) = \varphi(\boldsymbol{v}^\top \boldsymbol{x})$ for some nonlinear activation function $\varphi(\cdot)$. The prediction of the $k^{\text{th}}$ RFRR model is then given by

$$\hat{f}^k(\boldsymbol{x}) = \frac{\hat{\boldsymbol{w}}^{k\top} \boldsymbol{\psi}^k(\boldsymbol{x})}{\sqrt{N}}, \qquad (1)$$

where $\hat{\boldsymbol{w}}^k$ is the weight vector learned via ridge regression:

$$\hat{\boldsymbol{w}}^k = \arg\min_{\boldsymbol{w} \in \mathbb{R}^D} \left\{ \sum_{\mu=1}^P \left( y_\mu - \hat{f}^k(\boldsymbol{x}_\mu) \right)^2 + \lambda ||\boldsymbol{w}^k||_2^2 \right\} \quad (2)$$

$$= \left( \frac{1}{N} \boldsymbol{\Psi}^{k\top} \boldsymbol{\Psi}^k + \lambda \mathbf{I} \right)^{-1} \frac{\boldsymbol{\Psi}^{k\top} \boldsymbol{y}}{\sqrt{N}} \qquad (3)$$

where $\boldsymbol{\Psi}^k \in \mathbb{R}^{N \times P}$ has columns $[\boldsymbol{\psi}^k(\boldsymbol{x}_1), \cdots, \boldsymbol{\psi}^k(\boldsymbol{x}_P)]$ and the vector $\boldsymbol{y} \in \mathbb{R}^P$ has $[\boldsymbol{y}]_p = y_p$. Finally, the ensembled predictor is constructed by averaging the predictions of each ensemble member $k = 1, \ldots, K$:

$$\hat{f}_{\text{ens}}(\boldsymbol{x}) = \frac{1}{K} \sum_{k=1}^K \hat{f}^k(\boldsymbol{x}), \qquad (4)$$

**The Kernel Limit** Each of the learned functions $\hat{f}^k(\boldsymbol{x})$ may equivalently be expressed as the kernel ridge regression predictor:

$$\hat{f}^k(\boldsymbol{x}) = \hat{\boldsymbol{h}}_{x,\mathcal{X}}^k \left( \hat{\boldsymbol{H}}_{\mathcal{X}\mathcal{X}}^k + \lambda \boldsymbol{I}_N \right)^{-1} \boldsymbol{y} \qquad (5)$$

Where the matrix $[\hat{\boldsymbol{H}}_{\mathcal{X}\mathcal{X}}^k]_{pp'} = \hat{\boldsymbol{H}}^k(\boldsymbol{x}_p, \boldsymbol{x}_{p'})$ and the vector $[\hat{\boldsymbol{h}}_{x,\mathcal{X}}^k]_p = \hat{\boldsymbol{H}}^k(\boldsymbol{x}, \boldsymbol{x}_p)$, for the *stochastic* finite-feature kernels:

$$\hat{\boldsymbol{H}}^k(\boldsymbol{x}, \boldsymbol{x}') = \frac{1}{N} \sum_{n=1}^N g(\boldsymbol{v}_n^k, \boldsymbol{x}) g(\boldsymbol{v}_n^k, \boldsymbol{x}') \qquad (6)$$

The fluctuations in this stochastic kernel vary due to the random realization of the weights $\{\boldsymbol{v}_n^k\}_{n=1}^N$ across ensemble members $k = 1, \ldots, K$. As $N \to \infty$, these stochastic kernels converge to a common deterministic limiting kernel:

$$\hat{\boldsymbol{H}}^k(\boldsymbol{x}, \boldsymbol{x}') \xrightarrow{N \to \infty} \boldsymbol{H}(\boldsymbol{x}, \boldsymbol{x}') = \mathbb{E}_{\boldsymbol{v} \sim \mu_{\boldsymbol{v}}} [g(\boldsymbol{v}, \boldsymbol{x}) g(\boldsymbol{v}, \boldsymbol{x}')]$$

In this "kernel limit", the learned functions $\hat{f}^k(\boldsymbol{x})$ will be independent of the ensemble index $k$, so that we can define $\hat{f}_\infty(\boldsymbol{x}) \equiv \hat{f}_{ens}(\boldsymbol{x}) = \hat{f}^k(\boldsymbol{x}) \, \forall \, k$, with

$$\hat{f}_\infty(\boldsymbol{x}) = \boldsymbol{h}_{x,\mathcal{X}} (\boldsymbol{H}_{\mathcal{X}\mathcal{X}} + \lambda \boldsymbol{I})^{-1} \boldsymbol{y}, \quad (N \to \infty). \quad (7)$$

where $\boldsymbol{H}_{\mathcal{X}\mathcal{X}} \in \mathbb{R}^{P \times P}$ is the kernel matrix with entries $[\boldsymbol{H}_{\mathcal{X}\mathcal{X}}]_{pp'} = \boldsymbol{H}(\boldsymbol{x}_p, \boldsymbol{x}_{p'})$, and $\boldsymbol{h}_{x,\mathcal{X}} = [\boldsymbol{H}(\boldsymbol{x}, \boldsymbol{x}_1), \ldots, \boldsymbol{H}(\boldsymbol{x}, \boldsymbol{x}_P)]$. The limiting kernel $\boldsymbol{H}(\boldsymbol{x}, \boldsymbol{x}')$ can be decomposed into its eigenfunctions $\{\phi_t(\boldsymbol{x})\}_{t=1}^{\infty}$ and corresponding eigenvalues $\{\eta_t\}_{t=1}^{\infty}$:

$$\boldsymbol{H}(\boldsymbol{x}, \boldsymbol{x}') = \sum_{t=1}^{\infty} \eta_t \phi_t(\boldsymbol{x}) \phi_t(\boldsymbol{x}'). \tag{8}$$

In this formulation, $\hat{f}(\boldsymbol{x})$ is equivalent to the function learned by linear regression in the infinite-dimensional features with components given by $\theta_t(\boldsymbol{x}) \equiv \sqrt{\eta_t}\phi_t(\boldsymbol{x})$. We will assume that the target function can be decomposed in this basis as $f_*(\boldsymbol{x}) = \sum_t \bar{w}_t \theta_t(\boldsymbol{x})$. In the following sections, we will describe an error formula for RFRR ensembles which depends on these limiting kernel eigenvalues $\{\eta_t\}_{t=1}^{\infty}$ and target weights $\{w_t\}_{t=1}^{\infty}$, $K$, $N$, and $P$, which provide a complete statistical description of the learning problem.

**Gaussian Universality Assumption.** Following (Simon et al., 2023), we assume that the random features can be replaced by a Gaussian projection from the RKHS associated with the limiting kernel. Specifically, population risk is well described by the error formula obtained when the random features are replaced by $\boldsymbol{\psi}^k(\boldsymbol{x}) = \boldsymbol{Z}^k \boldsymbol{\theta}(\boldsymbol{x})$ where $[\boldsymbol{\theta}(\boldsymbol{x})]_t = \theta_t(\boldsymbol{x})$, $\boldsymbol{Z}^k \in \mathbb{R}^{N \times H}$ is a random Gaussian matrix with entries $Z_{ij}^k \overset{\text{i.i.d.}}{\sim} \mathcal{N}(0,1)$, and $H$ is the (infinite) dimensionality of the RKHS. Equivalently, we may approximate the stochastic kernels $\hat{\boldsymbol{H}}^k(\boldsymbol{x}, \boldsymbol{x}')$ as the inner products of linear random features:

$$\hat{\boldsymbol{H}}^k(\boldsymbol{x}, \boldsymbol{x}') = \frac{1}{N}\boldsymbol{\psi}^k(\boldsymbol{x})^\top \boldsymbol{\psi}^k(\boldsymbol{x}') = \frac{1}{N}\boldsymbol{\theta}(\boldsymbol{x})^\top \boldsymbol{Z}^{k\top} \boldsymbol{Z}^k \boldsymbol{\theta}(\boldsymbol{x}'). \tag{9}$$

Under this Gaussian universality assumption, the RFRR setting is equivalent to the linear random-features setting studied in ref's. (Maloney et al., 2022; Atanasov et al., 2024; 2022), with the limiting kernel eigenspectrum playing the role of the data covariance.

The Gaussian universality assumption has been justified rigorously in the case of a single ensemble member ($K = 1$) by Defilippis et al., who provide a multiplicative error bound on the resulting estimate for population risk (see section 3 and Appendix B).

**Test Risk** The test risk (also known as the generalization error or test error) quantifies the expected error of the learned function on unseen data. In this work, we define the test error as the mean squared error (MSE) between the predicted function $f_{\text{ens}}(\boldsymbol{x})$ and the true target function $f_*(\boldsymbol{x})$, averaged over the data distribution $\mu_{\boldsymbol{x}}$:

$$\mathcal{E}_g^K = \mathbb{E}_{\boldsymbol{x} \sim \mu_{\boldsymbol{x}}}\left[(\hat{f}_{\text{ens}}(\boldsymbol{x}) - f_*(\boldsymbol{x}))^2\right] + \sigma_\epsilon^2. \tag{10}$$

For binary classification problems, we might also consider the classification error rate on held-out test examples under score-averaging or a majority vote (equations A.6, A.7).

## 2.2. Degrees of Freedom

Using notation similar to (Atanasov et al., 2024) and (Bach, 2023), we will write expressions in terms of the "degrees of freedom" defined as follows:

$$\text{Df}_n(\kappa) \equiv \sum_t \frac{\eta_t^n}{(\eta_t + \kappa)^n}, \quad \text{tf}_n(\kappa) \equiv \sum_t \frac{\bar{w}_t^2 \eta_t^n}{(\eta_t + \kappa)^n}, \tag{11}$$

where $n \in \mathbb{N}$. Intuitively, $\text{Df}_n(\kappa)$ can be understood as a measure of how many modes of the kernel eigenspectrum are above a threshold $\kappa$, with the sharpness of the measurement increasing with $n$. $\text{tf}_n$ is a similar measure with each mode weighted by the corresponding component of the target function.

## 3. Deterministic Equivalent Error for Random Feature Ensembles

The error $\mathcal{E}_g^K$ of eq. 10 is a random quantity, depending both on the particular realization of the dataset $\mathcal{D}$ and the realization of the random feature vectors $\{\boldsymbol{v}_n^k\}$. Many recent works have found, however, that for large values of $P$ and $N$ the error $\mathcal{E}_g^K$ will concentrate over these sources of randomness. The error $\mathcal{E}_g^K$ can then be estimated by a *deterministic equivalent* error $E_g^K$, which depends on the dataset $\mathcal{D}$ and random features $\{\boldsymbol{v}_n^k\}$ only through their sizes $P$ and $N$. We will write $\mathcal{E}_g^K \simeq E_g^K$, with the symbol $\simeq$ meaning that the value on the right side is the deterministic quantity approximating the stochastic quantity on the left in the limit of large $P$ and $N$. The exact slack in this approximation varies in the literature, and is discussed in Appendix B.

We first review the deterministic equivalent approximation to the error $\mathcal{E}_g^1$ of a single RFRR model (superscript indicates $K = 1$). We do not derive this well-known result here, but rather direct the reader to a wealth of derivations, including references (Atanasov et al., 2024; Canatar et al., 2021; Simon et al., 2023; Adlam & Pennington, 2020; Rocks & Mehta, 2021; Hastie et al., 2022; Zavatone-Veth et al., 2022; Defilippis et al., 2024). Translating the risk estimate into our selected notation, we have that $\mathcal{E}_g^1 \simeq E_g^1$ with:

$$E_g^1 = \frac{1}{1 - \gamma_1}\left[-\rho\kappa_2^2 \text{tf}_1'(\kappa_2) + (1 - \rho)\kappa_2 \text{tf}_1(\kappa_2) + \sigma_\epsilon^2\right] \tag{12}$$

where we have defined

$$\rho \equiv \frac{N - \text{Df}_1(\kappa_2)}{N - \text{Df}_2(\kappa_2)}, \quad \gamma_1 \equiv \frac{1}{P}\left((1 - \rho)\text{Df}_1 + \rho\text{Df}_2\right) \tag{13}$$

and $\kappa_2$ is the solution to the following self-consistent equation:

$$\kappa_2 = \frac{\lambda N}{(P - \mathrm{Df}_1(\kappa_2))(N - \mathrm{Df}_1(\kappa_2))} \quad (14)$$

We find that eq. 12 provides an accurate estimate of risk at finite $N, P$.

### 3.1. The Bias-Variance Decomposition of $E_g$

The error formula can further be decomposed using a bias-variance decomposition with respect to the particular realization $\boldsymbol{Z}$ of random features:

$$E_g^1 = \mathrm{Bias}_z^2 + \mathrm{Var}_z \quad (15)$$

$$\mathrm{Bias}_z^2 \equiv \mathbb{E}_{\boldsymbol{x} \sim \mu_{\boldsymbol{x}}} \left[ \left( \mathbb{E}_{\boldsymbol{Z}} \left[ \hat{f}(\boldsymbol{x}) \right] - f_*(\boldsymbol{x}) \right)^2 \right] + \sigma_\epsilon^2 \quad (16)$$

$$\mathrm{Var}_z \equiv \mathbb{E}_{\boldsymbol{Z}} \mathbb{E}_{\boldsymbol{x} \sim \mu_{\boldsymbol{x}}} \left[ \left( \hat{f}(\boldsymbol{x}) - \mathbb{E}_{\boldsymbol{Z}} \left[ \hat{f}(\boldsymbol{x}) \right] \right)^2 \right] \quad (17)$$

While the learned function $f$ in the equation above also depends on the particular realization of the dataset $\mathcal{D}$, we do not include this in the Bias-Variance decomposition because we are interested in the variance due to the realization of a finite set of random features. Furthermore, the Bias and Variance written in equations 16, 17 are expected to concentrate over $\mathcal{D}$ (Atanasov et al., 2024; Adlam & Pennington, 2020; Lin & Dobriban, 2021). The deterministic equivalent formulas for the Bias and Variance are given explicitly in (Simon et al., 2023), and may be written as:

$$\mathrm{Bias}_z^2 \simeq \frac{-\kappa_2^2}{1 - \gamma_2} \mathrm{tf}_1'(\kappa_2) + \frac{\sigma_\epsilon^2}{1 - \gamma_2}, \quad (18)$$

$$\mathrm{Var}_z \simeq E_g^1 - \mathrm{Bias}_z^2, \quad (19)$$

where $\gamma_2 \equiv \frac{1}{P} \mathrm{Df}_2(\kappa_2)$.

### 3.2. Ensembling Reduces Variance of the Learned Estimator

Armed with a bias-variance decomposition of a single estimator over the realization of $\boldsymbol{Z}$, we can immediately write the risk estimate for an ensemble of $K$ estimators, each with an associated set of random features encapsulated by an independently drawn random Gaussian projection matrix $\boldsymbol{Z}^k$, $k = 1, \dots, K$. Because the realization $\boldsymbol{Z}^k$ of random features is the only parameter distinguishing the ensemble members, each ensemble member will have the same expected predictor $\mathbb{E}_{\boldsymbol{Z}} f^k(\boldsymbol{x})$. Furthermore, because the draws of $\boldsymbol{Z}^k$ are independent for $k = 1, \dots, K$, the deviations from this mean predictor will be independent across ensemble members, so that ensembling over $K$ predictors reduces the variance of the prediction by a factor of $K$:

$$E_g^K = \mathrm{Bias}_z^2 + \frac{1}{K} \mathrm{Var}_z \quad (20)$$

## 4. More is Better in Random Feature Ensembles

A ubiquitous observation in the modern practice of machine learning is that larger models and datasets lead to better performance. However, ensemble learning offers an additional complication that can break this trend. In the case of random forests, for example, subsampling of data dimensions leads to improved performance, despite *reducing* the expressivity of each decision tree (Breiman, 2001). Similarly, deep feature-learning ensembles have been empirically observed to violate the "bigger is better" principle (see fig. 8 from Vyas et al. (2023)). Here, we show that no such violations can occur in RFRR ensembles.

**Theorem 4.1.** *(More is better for RF Ensembles) Let $E_g^K(P, N, \lambda)$ denote $E_g^K$ with $P$ training samples, $N$ random features per ensemble member, ensemble size $K$, and ridge parameter $\lambda$ and any task eigenstructure $\{\eta_t\}_{t=1}^\infty$, $\{\bar{w}_t\}_{t=1}^\infty$, where $\{\eta_t\}_{t=1}^\infty$ has infinite rank. Let $K' \geq K$, $P' \geq P$ and $N' \geq N$. Then*

$$\min_\lambda E_g^{K'}(P', N', \lambda) \leq \min_\lambda E_g^K(P, N, \lambda) \quad (21)$$

*with strict inequality as long as $(K', N', P') \neq (K, N, P)$ and $\sum_t \bar{w}_t^2 \eta_t > 0$.*

Proof of this theorem follows from the omniscient risk estimate 20, and is provided in Appendix C. We make the following remarks:

*Remark* 4.2. In the special case $K = K' = 1$, this reduces to the "more is better" theorem for single models proven in Simon et al. (2023), and thus extends the notion that larger models and datasets improve performance from single models to RFRR ensembles.

*Remark* 4.3. Improved accuracy of the constituent models that form an ensemble does not necessarily lead to improved accuracy of the ensemble-averaged prediction, which suppresses the variance of the individual predictors. Proof of theorem 4.1 therefore requires a separate treatment of the bias and variance terms, unlike the $K = K' = 1$ case proven by Simon et al.

We demonstrate monotonicity with $P$ and $N$ in Fig. 1, where we plot $E_g^K$ as a function of both sample size $P$ and the network size $N$ in ensembles of ReLU random feature models applied to a binarized CIFAR-10 image classification task (see Appendix E.2). While error may increase with $P$ or $N$ at a particular ridge value $\lambda$, error decreases monotonically provided that the ridge $\lambda$ is tuned to its optimal value. Theoretical learning curves are calculated using eq. 20, with eigenvalues $\eta_k$ and target weights $\bar{w}_k$ determined by computing the NNGP kernel corresponding to the infinite-feature limit of the ReLU RF model (see Appendix E.3 for details). Numerically, we verify that error monotonicity with $P$ and $N$ holds at the level of a 0-1 loss on the

predicted classes of held-out test examples for both score-averaging and majority-vote ensembling over the predictors (see fig. S3).

We compare ridge-optimized error across ensemble sizes $K$ in fig. S1, finding that increasing sample size $P$ and network size $N$ are usually more effective than ensembling over multiple networks in reducing predictor error, indicating that for the binarized CIFAR-10 RFRR classification task, bias is the dominant contribution to error. Similarly, ensembling over multiple networks gives meager improvements in performance relative to increasing network size in deep feature-learning ensembles (Vyas et al., 2023).

# 5. No Free Lunch from Random Feature Ensembles

It is immediate from eq. 20 that increasing the size of an ensemble reduces the error. However, training additional models incurs a significant cost. With a fixed memory capacity, the machine learning practitioner is faced with the decision of whether to train a single large model, or to train an ensemble of smaller models. Here, we prove a "no free lunch" theorem which says that, given a fixed total number of features $M$ divided evenly among $K$ random feature models, then the lowest possible risk will always be achieved by $K = 1$, provided that the ridge is tuned to its optimal value. Furthermore, this ridge-optimized error increases monotonically with $K$.

**Theorem 5.1.** *(No Free Lunch From Random Feature Ensembles) Let $E_g^K(P, N, \lambda)$ denote $E_g^K$ with $P$ training samples, $N$ random features per ensemble member, ridge parameter $\lambda$, ensemble size $K$, and task eigenstructure $\{\eta_t\}_{t=1}^{\infty}$, $\{\bar{w}_t\}_{t=1}^{\infty}$, where $\{\eta_t\}_{t=1}^{\infty}$ has infinite rank. Let $K' < K$. Then*

$$\min_{\lambda} E_g^{K'}(P, M/K', \lambda) \leq \min_{\lambda} E_g^K(P, M/K, \lambda) \quad (22)$$

*with strict inequality as long as $\sum_t \bar{w}_t^2 \eta_t > 0$.*

Several remarks are in order:

*Remark* 5.2. In the limit where $M \to \infty$, this result implies that *no* ensemble of RFRR models can outperform the limiting (infinite-feature) KRR model with optimal ridge.

*Remark* 5.3. In section 7, we comment on the distinction between this result and known equivalences between infinite ensembles and single predictors of infinite size (Patil & LeJeune, 2024; LeJeune et al., 2020; Dern et al., 2024)

*Remark* 5.4. While ensembling is the canonical strategy for reducing the variance of a predictor, the proof of theorem 5.1 relies on the fact that when the ridge parameter is scaled to keep bias constant, increasing model size decreases variance of the learned predictor *faster* than adding additional predictors to the ensemble.

Proof of theorem 5.1 follows a similar strategy to the proof of theorem 4.1, and is provided in Appendix C. We test this prediction by performing ensembled ReLU RFRR on the binarized CIFAR-10 classification task in figure 2. We find that increasing $K$ while keeping the total number of features $M$ fixed always degrades the optimal test risk. The strength of this effect, however, depends on the size of the training set. For larger training sets ($P \gg N$), the width of each ensemble member becomes the constraining factor in each predictor's ability to recover the target function. However, when $P \ll N$, the optimal loss is primarily determined by $P$, so that optimal error only begins increasing appreciably with $K$ once $N = M/K \lesssim P$ (fig. 2 B). We again find similar behavior of the 0-1 loss under score-average and majority-vote ensembling (see fig. S6). While the ridge-optimized error is always minimal for $K = 1$, we notice in fig. 2C that near-optimal performance can be obtained with $K > 1$ over a wider range of $\lambda$ values, suggesting that ensembling may offer improved robustness in situations where fine-tuning of the ridge parameter is not possible. Further robustness benefits have been reported for regression ensembles of heterogeneous size (Ruben & Pehlevan, 2023).

Theorems 1 and 2 together guarantee that a larger ensemble of smaller RFRR ensembles can only outperform a smaller ensemble of larger RFRR models when the total parameter count of the former exceeds that of the latter. We formalize this fact in the following corollary:

**Corollary 5.5.** *Let $E_g^K(N)$ be the test risk of an ensemble of $K$ RFRR models each with $N$ features given by eq. 20. Suppose $K' > K$ or $N' < N$. It follows from Theorems 4.1 and 5.1 that*

$$\min_{\lambda} E_g^{K'}(N') \leq \min_{\lambda} E_g^K(N) \Rightarrow K'N' \geq KN. \quad (23)$$

*Remark* 5.6. We emphasize that this is a uni-directional statement. It is common for a single larger model to outperform an ensemble of smaller models with greater total size, particularly in the underparameterized regime. See, for example, fig. S1.B.

We demonstrate this result on synthetic tasks with power-law structure and on the binarized CIFAR-10 classification task in fig. S2.

# 6. Near-Optimality Conditions for RFRR Ensembles

Our results indicate that ensembling is always a sub-optimal allocation of model parameters. Theorem 5.1 guarantees that no ensemble of random-feature models can outperform a single model with the same total size when the ridge parameter is optimally tuned. Furthermore, corollary 5.5 states that an ensemble of smaller models can only outperform an ensemble of larger models if the ensemble of smaller

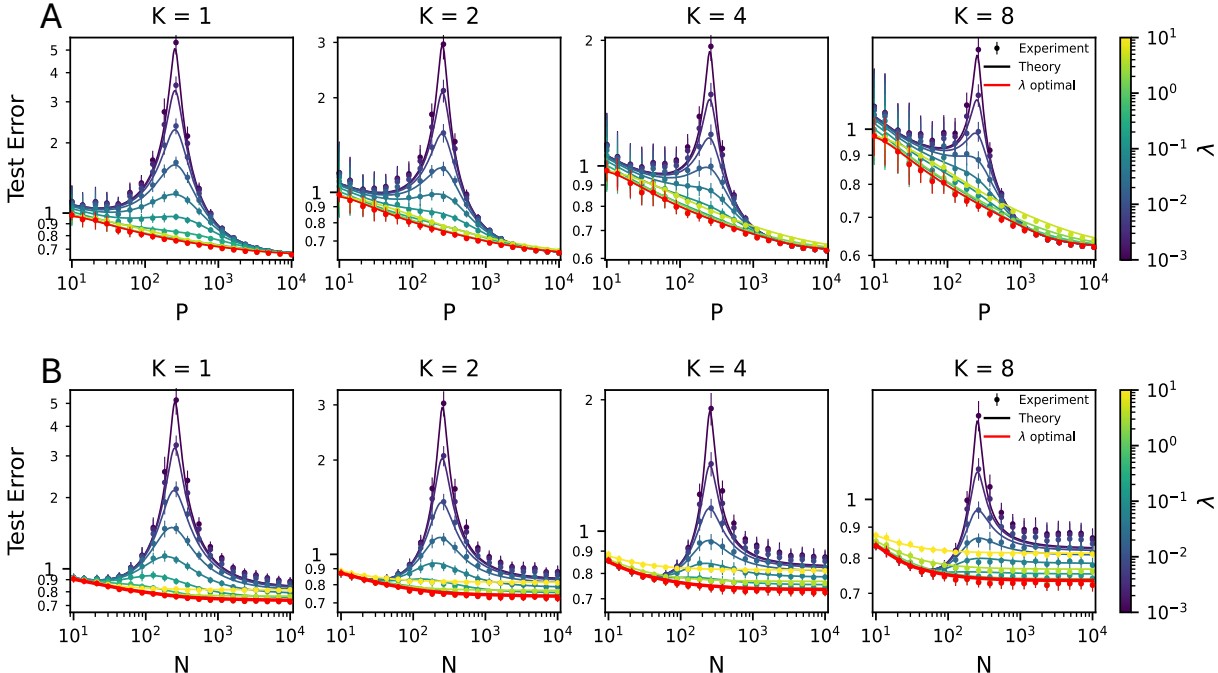

*Figure 1.* "More is better" in random feature ensembles. We perform ReLU RFRR on a binarized CIFAR-10 classification task and compare the empirical test risk to the omniscient risk estimate (eq. 20). (A) We fix $N = 256$ and vary both $P$ and $K$. Color corresponds to the regularization $\lambda$. Markers show numerical experiments and dotted lines theoretical predictions. Error is monotonically decreasing with $P$ provided that the regularization $\lambda$ is tuned to its optimal value. (B) Same as (A) except that $P = 256$ is fixed and $K$, $N$ are varied. Markers and error bars show mean and standard deviation over 50 trials.

models has greater total size. These results suggest that, with a fixed memory capacity, it is always beneficial to scale a predictor by increasing the size of the network. However, ensembling remains an attractive option in practice because they allow fully parallelized training and inference. It is therefore important to ask when ensembles can attain *near-optimal* performance. In section 6.1, we investigate RFRR ensembles in the overparameterized regime, finding that at leading order in $\lambda$ and $1/N$, error depends only on the total feature count $M = KN$. To study the underparameterized regime, we derive scaling laws for RFRR ensembles under joint scaling of ensemble size $K$ and model size $N$. While the optimal scaling law is always achieved by scaling model size with a fixed ensemble size, our analysis provides conditions on the data and task eigen-structure under which joint scaling of model size and ensemble size can achieve a near-optimal scaling law for the error.

### 6.1. Near-Optimal Risk in Overparameterized RFRR Ensembles

For ensembles in the overparameterized regime ($N \gg P$), the bound of corollary 5.5 appears to be tight, meaning that an ensemble of $K$ models of size $N$ performs similarly to a single model of size $M = NK$. To understand why, we

expand the deterministic equivalent error formula $E_g^K$ (eq. 20) as power series in $1/N \gtrsim 0$ and $\lambda \approx 0$ in Appendix D.2, finding:

$$
\begin{aligned}
E_g^K = &-\frac{P\kappa_2^{*\,2}\,\mathrm{tf}_1'(\kappa_2^*)}{P - \mathrm{Df}_2(\kappa_2^*)} + \lambda F(\kappa_2^*, P) + \frac{P\kappa_2^*\,\mathrm{tf}_1(\kappa_2^*)}{KN} \\
&+ \mathcal{O}(\lambda^2, \lambda/N, 1/N^2),
\end{aligned} \tag{24}
$$

where $\kappa_2^*$ satisfies $\mathrm{Df}_1(\kappa_2^*) = P$, and we have set $\sigma_\epsilon = 0$. We see that, at leading order in $\lambda$, and $1/N$, risk depends on the ensemble size $K$ and model size $N$ only through the total number of features $KN$. So, while Theorem 5.1 guarantees that there can be no benefit to dividing features into an ensemble of smaller predictors, we don't expect this to appreciably harm performance as long as the ensemble remains in the overparameterized regime ($N \gg P$) and optimal or near-optimal performance can be achieved with a small ridge parameter. While we do not provide exact conditions under which near-optimal performance is achieved at small ridge parameter here, our numerical experiments verify that error depends on $N$ and $K$ only through their product $M = NK$ (see agreement between dotted lines and green lines in fig. S2). Furthermore, Simon et al. argue that in the overparameterized limit $N \to \infty$, optimal ridge approaches zero for tasks with power-law structure under realistic conditions.

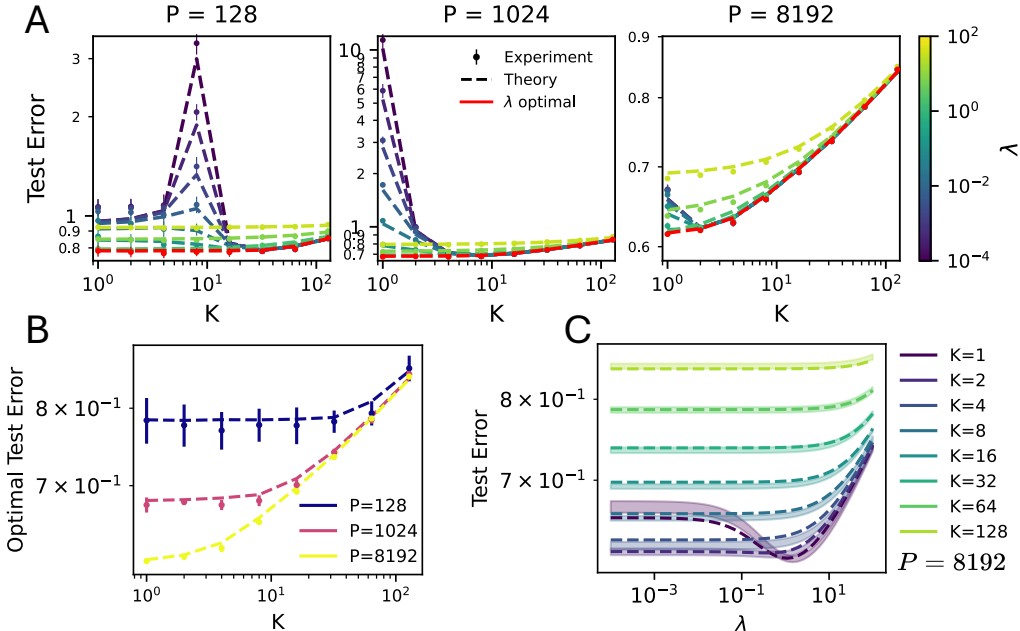

*Figure 2.* No Free Lunch from Random Feature Ensembles. We perform kernel RF regression on a binarized CIFAR 10 classification task. (A) We vary $K$ and $N$ while keeping total parameter count $M = 1024$ fixed. The sample size $P$ is indicated above each plot. (B) Error $E_g^K$ optimized over the ridge parameter $\lambda$ increases monotonically with $K$ provided the total parameter count $M$ is fixed. Dashed lines show theoretical prediction using eq. 20 and markers and error-bars show mean and standard deviation of the risk measured in numerical simulations across 10 trials. (C) We show error as a function of $\lambda$ for each $K$ value simulated and $P = 8192$. Dashed lines show theoretical prediction using eq. 20 and shaded regions show standard deviation of risk measured in numerical simulations across 10 trials.

## 6.2. Scaling Laws for Underparameterized RFRR Ensembles

We now study the behavior of RFRR ensembles in the underparameterized regime where $N = M/K \ll P$. Here, model size $N$ is the "bottleneck" that dictates the scaling behavior of the risk (Bahri et al., 2024; Maloney et al., 2022; Bordelon et al., 2024a; Atanasov et al., 2024). In particular, we ask how $E_g^K$ scales with total model size $M$ when $K$ and $N$ scale jointly with $M$ according to a "growth exponent" $\ell$:

$$\ell \in [0,1] \qquad K \sim M^{1-\ell} \qquad N \sim M^\ell, \qquad (25)$$

so that when $\ell = 0$ the ensemble grows with $M$ by adding additional ensemble members of a fixed size, and when $\ell = 1$ the ensemble grows by adding parameters to a fixed number of networks. Under the standard "source" and "capacity" constraints on the task eigenstructure (Cui et al., 2023; Caponnetto & De Vito, 2006; Bordelon et al., 2020; Defilippis et al., 2024) (the kernel eigenspectrum decays as $\eta_t \sim t^{-\alpha}$ with $\alpha > 1$ and the target weights decay as

$\bar{w}_t^2 \eta_t \sim t^{-(1+2\alpha r)}$), we find:

$$E_g^K \sim M^{-s} \qquad (26)$$

$$s = \min\left(2\alpha\ell\min(r,1), 1 - \ell + 2\alpha\ell\min\left(r, \frac{1}{2}\right)\right), \qquad (27)$$

with $s = 2\alpha\ell\min(r,1)$ corresponding to the scaling of the bias and $s = 1 - \ell + 2\alpha\ell\min(r, 1/2)$ corresponding the scaling of the variance (reduced by a factor of $1/K$). A full derivation is provided in Appendix D. These results reify the "no free lunch" result, as the optimal scaling law is always achieved when $\ell = 1$. For difficult tasks, defined as having $r < 1/2$, bias always dominates the error scaling and the scaling exponent increases linearly with $\ell$. However, when $r > 1/2$, there will be a certain value $\ell^*$ above which error scaling is dominated by the variance term. When $r > 1/2$, the scaling exponent of the variance increases from 1 to $\alpha$ over the range $\ell \in [0,1]$. If $\alpha \gtrsim 1$, this can approach a flat line, and the dependence of the scaling exponent on $\ell$ can become weak, so that *near-optimal* scaling can be achieved for any $\ell > \ell^*$. When $1/2 < r < 1$, this transition occurs at $\ell^* = 1/(1 + \alpha(2r - 1))$ and when $r > 1$ it occurs at $\ell^* = 1/(1 + \alpha)$.

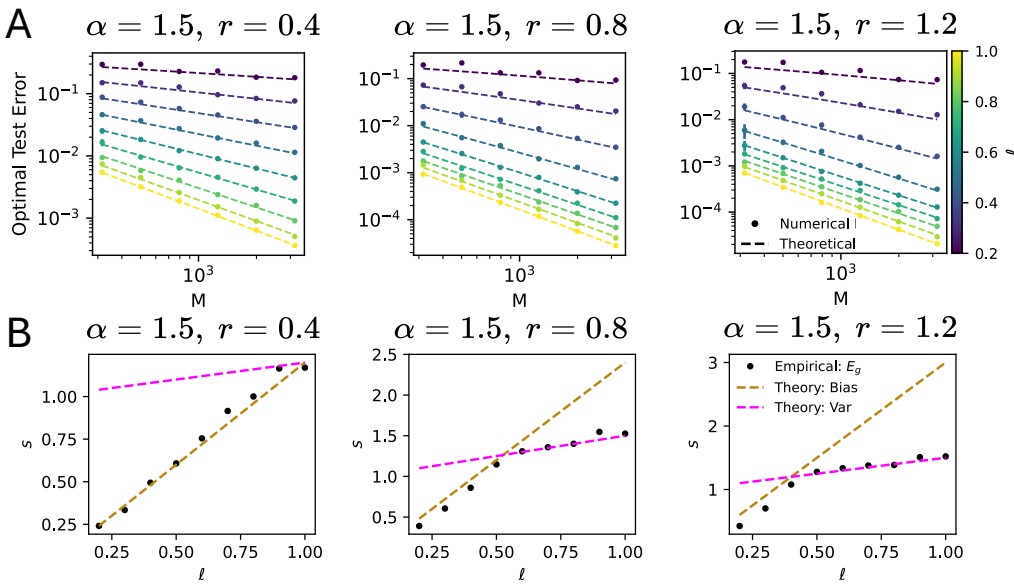

*Figure 3.* width-bottlenecked scaling laws of kernel RF regression under source and capacity constraints. We fix $P = 15,000$, $\alpha = 1.5$, and $r \in \{0.4, 0.8, 1.2\}$ and calculate $E_g^K$ as a function of $M$ with $N = M^\ell$ and $K = M^{(1-\ell)}$ using both the omniscient risk estimate (eq. 20) and numerical simulation of a linear Gaussian random-feature model (eq. A.12). (A) Plots of $E_g^K$ vs. $M$ at different $\ell$ values reveal that $\ell$ controls the scaling law of the error. (B) We plot the theoretical scaling exponents (eq. 26): Bias $\sim 2\alpha\ell \min(r, 1)$, Var $\sim 1 - \ell + 2\alpha\ell \min(r, \frac{1}{2})$ along with the scaling laws obtained by fitting the risks obtained by numerical simulation.

We plot these scaling laws with $\alpha = 1.5$ in the regimes where $r < 1/2$, $1/2 < r < 1$, and $r > 1$ in fig. 3, along with the results of numerical simulations of linear RF regression on synthetic Gaussian datasets. As anticipated, for difficult tasks, where $r < 1/2$, the scaling law improves linearly with $\ell$. However, for easier tasks ($r > 1/2$), we see that, a near-optimal scaling law can be achieved as long as $\ell > \ell^*$ (fig. 3, center and right columns).

We also determine the scaling behavior of the ReLU RFRR ensembles on the binarized CIFAR-10 and MNIST classification tasks. For both tasks, we calculate the statistics of the limiting kernel eigenspectrum $\{\eta_t\}_{t=1}^\infty$ and target weights $\{\bar{w}_t\}_{t=1}^\infty$ and fit their spectral decays to the source and capacity constraints. We find that for CIFAR-10, $\alpha \approx 1.33$, $r \approx 0.038$ and for MNIST $\alpha \approx 1.46$, $r \approx 0.14$, which places both tasks squarely in the difficult regime with $r < 1/2$. In figure 4, we show that the predicted scaling exponents of eq. 26 agree with empirical scaling laws for RFRR ensembles applied to the binarized CIFAR-10 and MNIST classification tasks.

## 7. Discussion

Our results show that in RFRR ensembles any violations of the "more is better" or "no free lunch from ensembles" principles are a result of overfitting at non-optimal ridge. This suggests that violations of these principles in other model types may also be a result of overfitting. Because deep networks reduce to kernel methods in the lazy training limit (Chizat et al., 2019; Jacot et al., 2018), we may pinpoint any violations of these principles in deep ensembles as a consequence specifically of feature learning. Future work should extend recent analytical treatments of feature-learning networks (Bordelon et al., 2024b) to the ensembled case.

Our results relate to a line of work on "sketched" or subsampled ridge ensembles, which prove that an *infinite* ensemble of regression models trained on random projections of the data achieve equivalent test risk to a single regression model trained on the full data, with an increased ridge parameter (Patil & LeJeune, 2024; Atanasov et al., 2024; LeJeune et al., 2020; Patil et al., 2023). Translated to the kernel eigenframework under Gaussian universality, these results imply that in the limit $K \to \infty$, RFRR ensembles converge on the same test risk as the limiting (infinite-feature) kernel predictor. It follows that *no* ensemble of RFRR models can outperform the limiting kernel predictor at optimal ridge, corresponding to the $M \to \infty$ limit of theorem 5.1 with $K' = 1$. Because it applies also at finite $M$ and $K' > 1$, theorem 5.1 is a stronger statement.

Our study also connects to recent work on scaling laws in deep learning (Kaplan et al., 2020; Hoffmann et al., 2022; Bordelon et al., 2024a;b), which observe that the test error of neural networks tends to improve predictably as a power-law

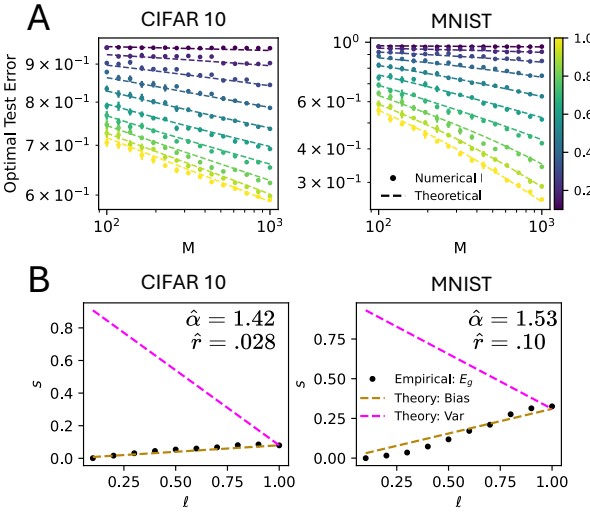

*Figure 4.* Scaling laws provide a good description of width-bottlenecked RFRR ensembles.(A) we plot error as a function of $M$ at optimal ridge value for $\mathrm{ReLU}$ random-feature models applied to the binarized CIFAR-10 (left) and MNIST (right) classification tasks. (B) We plot theoretically predicted scaling exponents (eq. 26) for the bias and variance contributions to risk, as well as empirical power-law fits to risk in numerical simulations of RFRR models (see Appendix E.3, fig. S7)

with the number of parameters and the size of the dataset used during training. With our scaling-law analysis, we extend the power-laws predicted using random-feature models (Bahri et al., 2024; Maloney et al., 2022; Bordelon et al., 2020; 2024a; Defilippis et al., 2024) to the case where model size $N$ and ensemble size $K$ are scaled jointly according to a novel "growth exponent" $\ell$ defined in eq. 25. While optimal scaling is always achieved by fixing $K$ and scaling $N$, for sufficiently easy tasks ($r > 1/2$), near-optimal scaling laws can be achieved by growing both $N$ and $K$, provided $N$ grows *quickly enough* with total parameter count. Because feature-learning networks can dynamically align their representations to the target function (Bordelon et al., 2024b), the scaling laws for deep ensembles may be dramatically improved by feature-learning effects.

An important limitation of our work is the assumption of statistically homogeneous ensembles. We consider each ensemble member to be trained on the same dataset, and to perform the same task. However, successes have been achieved using ensembles with *functional specialization*, where different sub-networks are trained on different datasets to perform different sub-tasks relevant to the overall goal of the ensemble. For example, mixture of experts (MoE) models (Jacobs et al., 1991; Lepikhin et al., 2020; Fedus et al., 2022) might offer a way to cleverly scale model size using ensembles that outperforms the scaling laws for single large networks. We leave a theory of ensembled regression which allows for

functional specialization as an objective for future work.

## 8. Conclusion

In this work, we analyzed a trade-off between ensemble size and features-per-ensemble-member in the tractable setting of RFRR. We prove a "no free lunch from ensembles" theorem which states that optimal performance is always achieved by allocating all features to a single, large RFRR model, provided that the ridge parameter is tuned to its optimal value. A scaling-laws analysis reveals that the sharpness of this trade-off depends sensitively on the structure of the task. In particular, near-optimal scaling laws can be achieved by RFRR ensembles, provided the task is sufficiently aligned with the top modes of the limiting kernel eigenspectrum. Our results tell a story consistent with the general trend from massively ensembled predictors to large models trained by joint optimization of all parameters with respect to a single loss function. Our results have implications for model design and resource allocation in practical settings, where model size is limited.

## Acknowledgements

C.P. is supported by NSF grant DMS-2134157, NSF CA-REER Award IIS-2239780, DARPA grant DIAL-FP-038, a Sloan Research Fellowship, and The William F. Milton Fund from Harvard University. WLT is supported by a Kempner Graduate Fellowship. HC was supported by the GFSD Fellowship, Harvard GSAS Prize Fellowship, and Harvard James Mills Peirce Fellowship. This work has been made possible in part by a gift from the Chan Zuckerberg Initiative Foundation to establish the Kempner Institute for the Study of Natural and Artificial Intelligence. BSR thanks Blake Bordelon, Alex Atanasov, Sabarish Sainathan, James B. Simon, and especially Jacob Zavatone-Veth for thoughtful discussion related to and comments on this manuscript.

## Impact Statement

This paper presents work whose goal is to advance the field of Machine Learning. There are many potential societal consequences of our work, none which we feel must be specifically highlighted here.

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

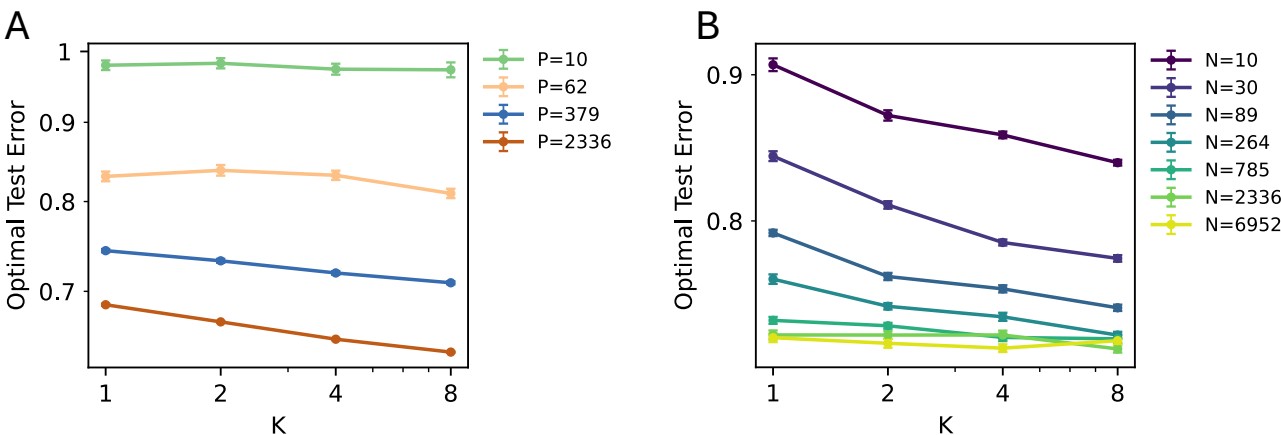

*Figure S1.* $E_g^k$ at optimal ridge as a function of ensemble size $K$ for binarized CIFAR-10 RFRR classification. (A) We fix $N = 256$, $P$ values indicated in legend. (B) We fix $P = 256$, $N$ values indicated in legend. Error bars show standard deviation across 10 trials.

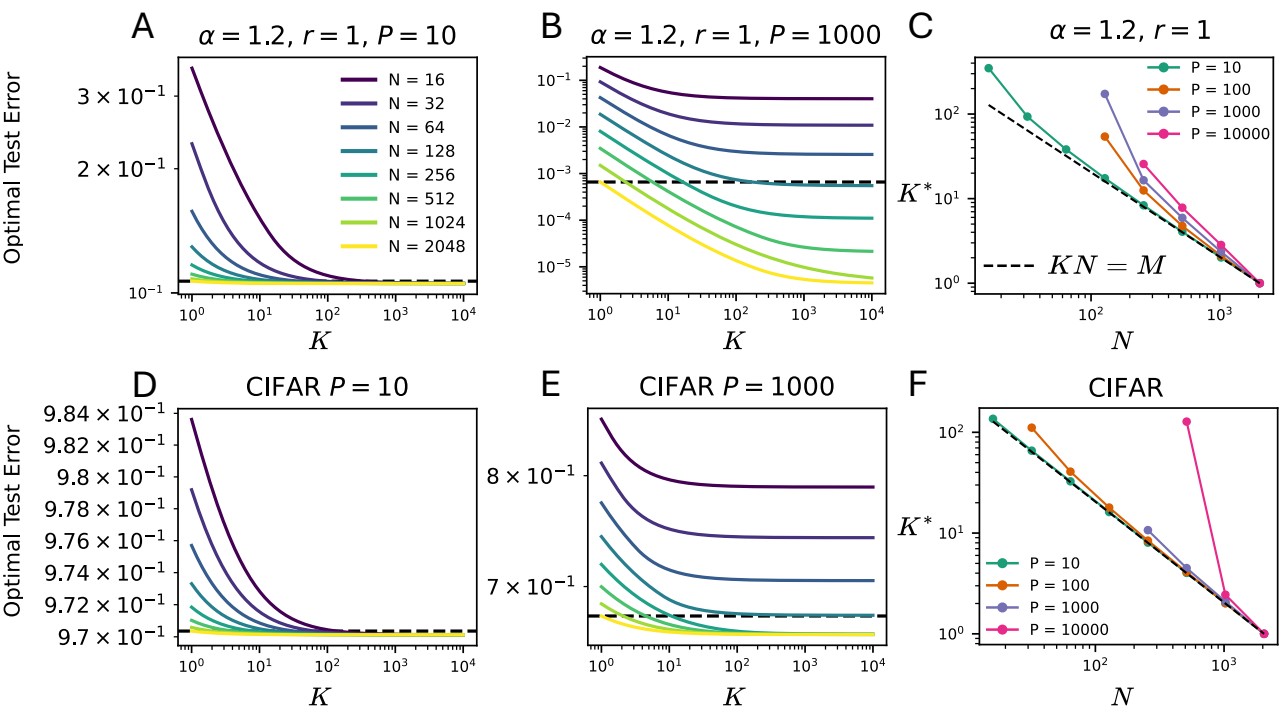

*Figure S2.* (A, B, D, E) We plot theoretical values for $E_g^k$ at optimal ridge as a function of ensemble size $K$ for RFRR with power-law eigenstructure with source exponent $r = 1$ and capacity $\alpha = 1.2$ (A, B) and for the NNGP kernel associated with the binarized CIFAR-10 classification task (D, E) . Random features per ensemble member $N$ shown in the legend. The dotted black line shows $E_g^1$ for a single RFRR model with $N = M = 2048$ features. Sample size $P$ is indicated in the title. (C, F) We plot the ensemble size $K^*$ for which an ensemble of RFRR models with size $N$ performs at least as well as single RFRR model with $M = 2048$ random features. $P$ values indicated in legend. As predicted by corollary 5.5, all curves lie above the dotted line $KN = M$. This bound appears tight when $P \ll N$.

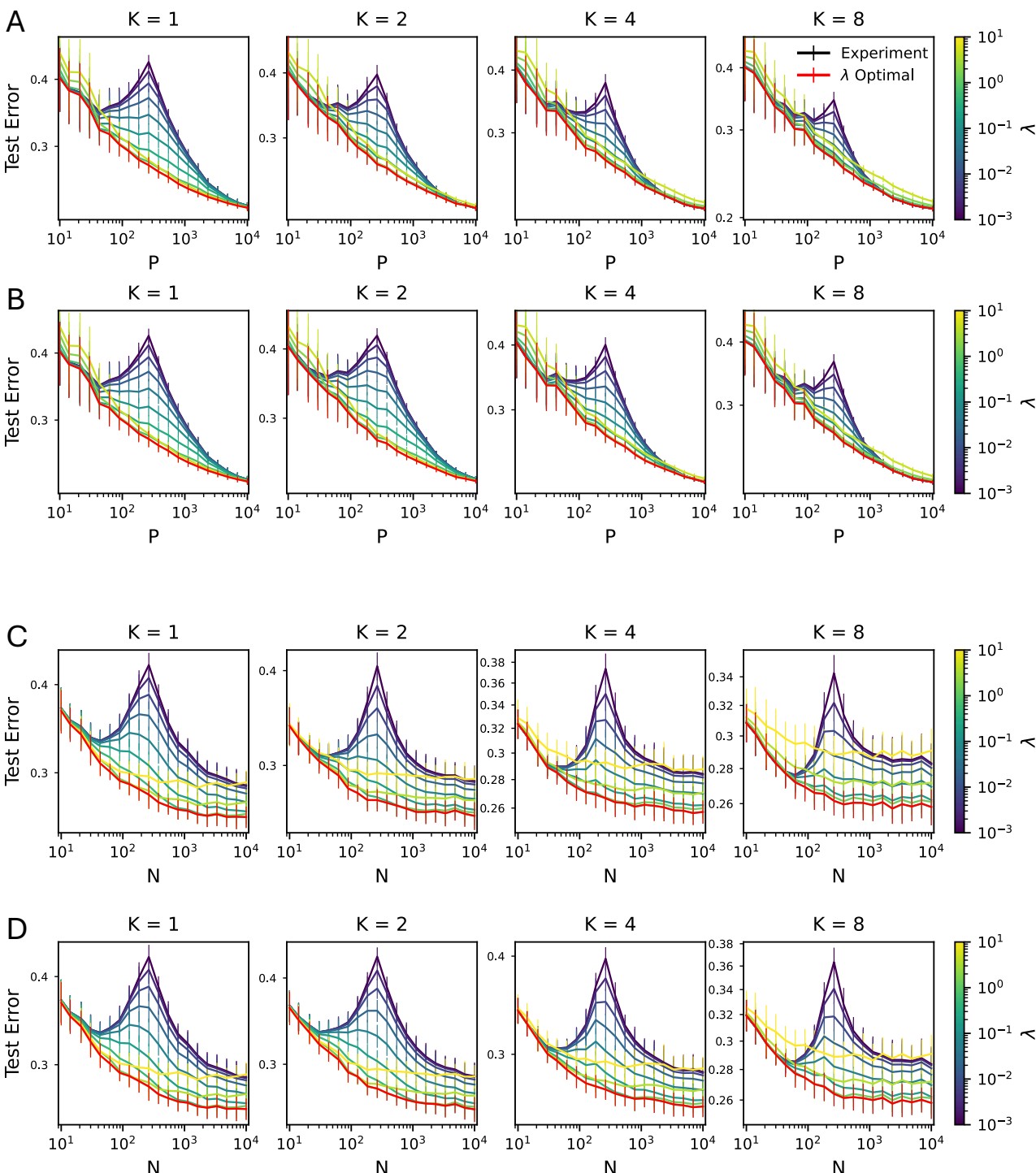

*Figure S3.* 0-1 Loss for binarized CIFAR-10 RFRR task under score-averaging (A, C) and majority vote (B, D) ensembling schemes. As in Fig. 1, Errors are shown (A, B) as a function of $P$ for fixed $N = 256$ and (B) as a function of $N$ for fixed $P = 256$. $K$ value indicated in title and $\lambda$ value in colorbar. Red line indicates optimal ridge determined by grid search. Markers and error bars show mean and standard deviation over 50 trials.

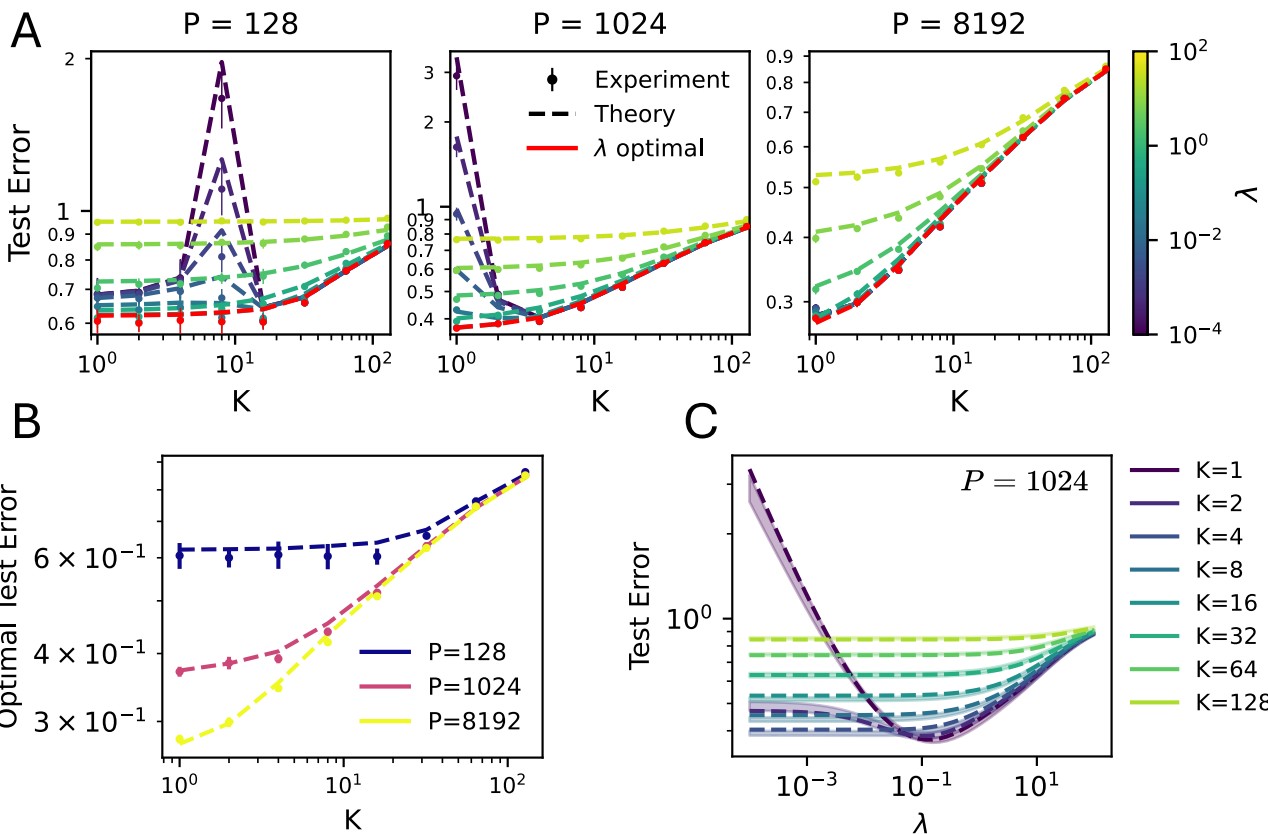

*Figure S4.* No Free Lunch from Ensembles of Random Feature Models. $E_g$ for kernel RF regression on an MNIST classification task. (A) Warying $K$ and $N$ while keeping total parameter count $M = 1024$ fixed. The sample size $P$ is indicated above each plot. (B) Error $E_g^K$ optimized over the ridge parameter $\lambda$ increases monotonically with $K$ provided the total parameter count $M$ is fixed. Dashed lines show theoretical prediction using eq. 20 and markers and error-bars show mean and standard deviation of the risk measured in numerical simulations across 10 trials. (C) We show error as a function of $\lambda$ for each $K$ value simulated and $P = 8192$. Dashed lines show theoretical prediction using eq. 20 and shaded regions show standard deviation of risk measured in numerical simulations across 10 trials.

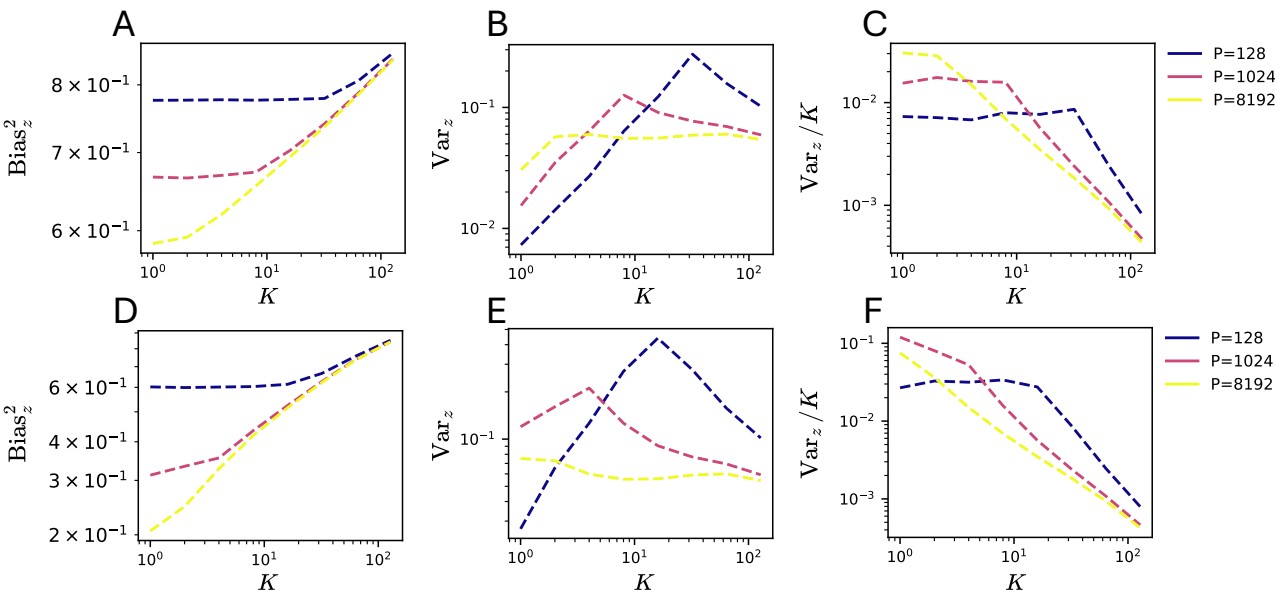

*Figure S5.* Bias-Variance decomposition of error at optimal ridge for binarized CIFAR-10 (A, B, C) and MNIST (D, E, F) RFRR tasks. We vary $K$ and $N$ while keeping total parameter $M = 1024$ fixed. $\text{Bias}_z^2$ (A, D), single-predictor variance $\text{Var}_z$ (B, E), and ensemble-predictor variance $\text{Var}_z / K$ (C, F) are calculated from theoretical expressions 19.

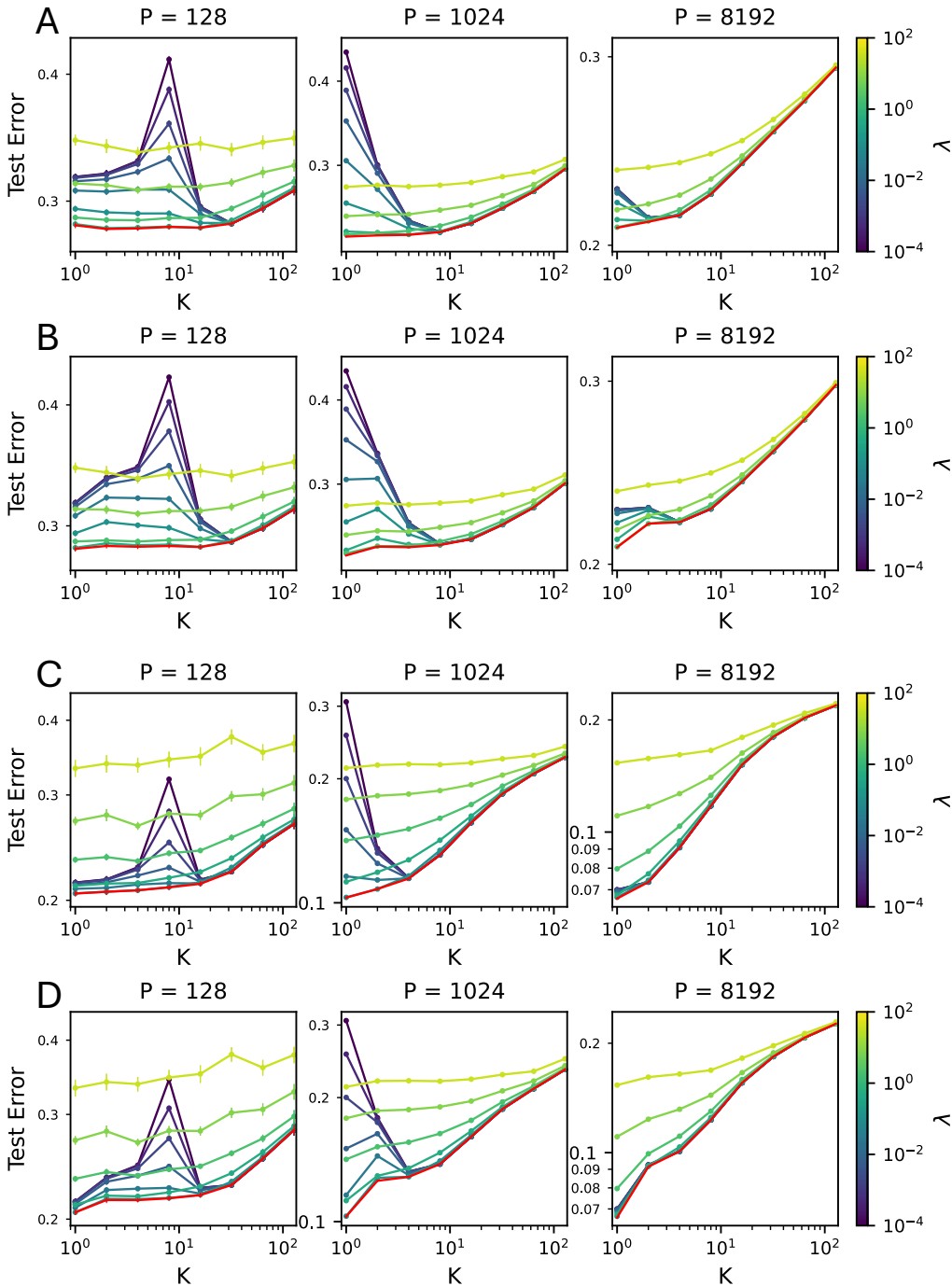

*Figure S6.* 0-1 loss for binarized CIFAR-10 (A, B) and MNIST (C, D) RFRR classification tasks under score-average (A, C) and majority vote (B, D) ensembling. We sweep $K$ and $N$ keeping $M = KN = 1024$ fixed. Sample size $P$ indicated in titles. Colorbar indicates ridge parameter. Red indicates optimal ridge determined by grid search.

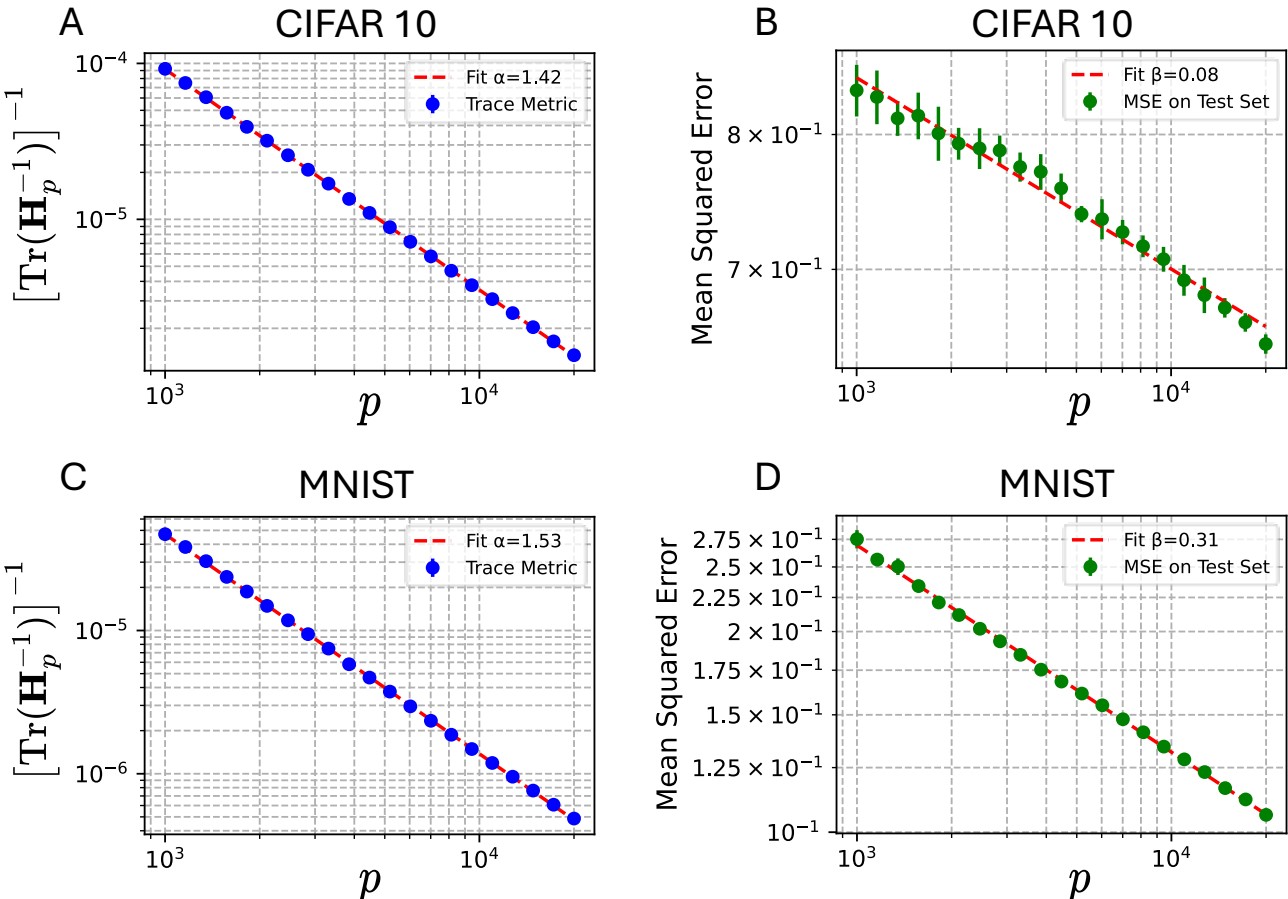

*Figure S7.* We measure the eigenspectrum of the NNGP kernel applied to the CIFAR-10 and MNIST datasets, as well as the target weights for the binarized classification tasks described in Appendix E.2. Estimates for the source and capacity exponents are obtained by fitting the "trace metric" $\left[\operatorname{tr}\left[\boldsymbol{H}_p\right]^{-1}\right]^{-1}$ and the MSE loss of kernel ridge regression with the limiting NNGP kernel to power laws (see Appendix E.4 for details).

## A. Extending the Random-Feature Eigenframework to Ensembles

We consider the RFRR setting as described by (Simon et al., 2023), extended to ensembles of predictors. A detailed description of this extended framework is Let $\mathcal{D} = \{\boldsymbol{x}_p, y_p\}_{i=1}^P$ be a training set of $P$ examples, where $\boldsymbol{x}_p \in \mathbb{R}^D$ are input features and $y_p \in \mathbb{R}$ are target values generated by a noisy ground-truth function $y_p = f_*(\boldsymbol{x}_p) + \epsilon_p$ and the label noise $\epsilon_p \overset{\text{i.i.d.}}{\sim} \mathcal{N}(0, \sigma_\epsilon^2)$.

We consider an ensemble of $K$ random feature models, each with $N$ features. The total number of features is thus $M = K \cdot N$. For each model $k = 1, \ldots, K$, we sample $N$ feature vectors $\{\boldsymbol{w}_n^k\}_{n=1}^N$ i.i.d. from a measure $\mu_{\boldsymbol{v}}$ over $\mathbb{R}^B$ (we will use upper indices to represent the index of the ensemble member, and lower indices to represent indices of the training examples and features). An ensemble of $K$ *featurization transformations* are defined as $\psi^k : \boldsymbol{x} \mapsto (g(\boldsymbol{v}_n^k, \boldsymbol{x}))_{n=1}^N$ where $g : \mathbb{R}^B \times \mathbb{R}^D \to \mathbb{R}$ is square-integrable with respect to $\mu_{\boldsymbol{x}}$ and $\mu_{\boldsymbol{v}}$. The predictions of the ridge regression models are then given as $f^k(\boldsymbol{x}) = \boldsymbol{w}^k \cdot \psi(\boldsymbol{x})/\sqrt{N}$, where the weight vectors $\boldsymbol{w}^k$ are determined by standard linear ridge regression with a ridge parameter $\lambda$:

$$\hat{\boldsymbol{w}}^k = \left(\frac{1}{N} \boldsymbol{\Psi}^{k\top} \boldsymbol{\Psi}^k + \lambda \mathbf{I}\right)^{-1} \frac{\boldsymbol{\Psi}^{k\top} \boldsymbol{y}}{\sqrt{N}} \tag{A.1}$$

Where the matrices $\boldsymbol{\Psi}^k \in \mathbb{R}^{N \times P}$ have columns $[\psi^k(\boldsymbol{x}_1), \cdots, \psi^k(\boldsymbol{x}_P)]$ and the vector $\boldsymbol{y} \in \mathbb{R}^P$ has $[\boldsymbol{y}]_p = y_p$. For each ensemble member, this is equivalent to the kernel ridge regression predictor:

$$f^k(\boldsymbol{x}) = \hat{\boldsymbol{h}}_{x,\mathcal{X}} \left(\hat{\boldsymbol{H}}_{\mathcal{X}\mathcal{X}} + \lambda \boldsymbol{I}_N\right)^{-1} \boldsymbol{y} \tag{A.2}$$

Where the matrix $[\hat{\boldsymbol{H}}_{\mathcal{X}\mathcal{X}}]_{pp'} = \hat{\boldsymbol{H}}^k(\boldsymbol{x}_p, \boldsymbol{x}_{p'})$ and the vector $[\hat{\boldsymbol{h}}_{x,\mathcal{X}}]_p = \hat{\boldsymbol{H}}^k(\boldsymbol{x}, \boldsymbol{x}_p)$ with the stochastic finite-feature kernel

$$\hat{\boldsymbol{H}}^k(\boldsymbol{x}, \boldsymbol{x}') = \frac{1}{N} \sum_{n=1}^N g(\boldsymbol{w}_n^k, \boldsymbol{x}) g(\boldsymbol{w}_n^k, \boldsymbol{x}') \qquad k = 1, \ldots, K \tag{A.3}$$

In the limit of infinite features, this stochastic kernel converges to a deterministic limit $\hat{\boldsymbol{H}}^k(\boldsymbol{x}, \boldsymbol{x}') \to \boldsymbol{H}(\boldsymbol{x}, \boldsymbol{x}')$. Because we consider the feature function $g$ to be shared across ensemble members, this limit is independent of $k$. The ensemble prediction is the average of individual model predictions:

$$f_{\text{ens}}(\boldsymbol{x}) = \frac{1}{K} \sum_{k=1}^K f^k(\boldsymbol{x}) \tag{A.4}$$

Finally, we measure the test error as the mean-squared error of the ensemble as the mean-squared error on a held out-test sample:

$$E_g^K \equiv \mathbb{E}_{\boldsymbol{x} \sim \mu_{\boldsymbol{x}}} \left[(f_{\text{ens}}(\boldsymbol{x}) - f_*(\boldsymbol{x}))^2\right] + \sigma_\epsilon^2 \tag{A.5}$$

For binary classification problems, we may be more interested in the classification error rate for the learned predictor. Given an ensemble of scalar output "scores" $f^1(\boldsymbol{x}), \ldots, f^K\boldsymbol{x}$, two possible schemes to assign the class of the test example $\boldsymbol{x}$ are score-averaging and majority-vote ensembling (Loureiro et al., 2022):

$$f_{\text{ens}}^{\text{SA}}(\boldsymbol{x}) = \text{Sign}\left(\sum_{k=1}^K f^k(\boldsymbol{x})\right) \qquad \text{(Score-Average)} \tag{A.6}$$

$$f_{\text{ens}}^{\text{MV}}(\boldsymbol{x}) = \text{Sign}\left(\sum_{k=1}^K \text{Sign}\left(f^k(\boldsymbol{x})\right)\right) \qquad \text{(Majority-Vote)} \tag{A.7}$$

The classification error rate is then given as the probability of mislabeling a held-out test example.

### A.1. Spectral Decomposition of the Kernel

The feature function $g$ permits a spectral decomposition as follows:. Let $T : L^2(\mu_{\boldsymbol{v}}) \to L^2(\mu_{\boldsymbol{x}})$ be the linear operator defined by:

$$(Tr)(x) = \int_{\mathbb{R}^B} r(\boldsymbol{v}) g(\boldsymbol{v}, \boldsymbol{x}) d\mu_{\boldsymbol{v}}(\boldsymbol{v}) \tag{A.8}$$

The singular value decomposition of $T$ (Kato, 1966) yields orthonormal bases $\{\zeta_n\}_{n=1}^{\infty}$ of $\mathrm{Ker}^{\perp}(T) \subset L^2(\mu_v)$ and $\{\phi_n\}_{n=1}^{\infty}$ of $L^2(\mu_n)$, where $\{\eta_t\}_{t=1}^{\infty}$ are the eigenvalues (in decreasing order) and $\{\zeta_t\}_{t=1}^{\infty}$ the corresponding eigenvectors integral operator $\Sigma : L^2(\mu_x) \to L^2(\mu_x)$ given by

$$(\Sigma u)(x) = \int_{\mathbb{R}^D} u(x') H(x', x) d\mu_x(x') \tag{A.9}$$

We can write $\Sigma = TT^{\star}$, where $T^{\star} : L^2(\mu_x) \mapsto L^2(\mu_v)$ denotes the adjoint of $T$. The feature function $g$ can then be decomposed as $g(v, x) = \sum_{t=1}^{\infty} \sqrt{\eta_t} \zeta_t(v) \phi_t(x)$.

Under these conditions, we may write the stochastic finite-feature kernel functions as:

$$\hat{H}^k(x, x') = \frac{1}{N} \sum_{n=1}^{N} \sum_{t, t'=1}^{\infty} \sqrt{\eta_t \eta_{t'}} \zeta_t(v_n^k) \zeta_{t'}(v_n^k) \phi_t(x) \phi_t'(x') \tag{A.10}$$

Using the orthonormality of the bases, the deterministic limit of the kernel function can then be expanded as

$$H(x, x') = \sum_t \eta_t \phi_t(x) \phi_t(x') = \sum_t \theta_t(x) \theta_t(x') \tag{A.11}$$

where we have defined $\theta_t(x) \equiv \sqrt{\eta_t} \phi_t(x)$. We see that that the singular values $\{\eta_t\}$ of the operator $T$ double as the eigenvalues of the limiting kernel operator $H$. We will assume that the ground truth function $f_*(x)$ can then be decomposed as: $f_*(x) = \sum_t \bar{w}_t \theta_t(x)$. Any component of $f_*$ which does not lie in the RKHS of the kernel could, in principle, be absorbed into the noise $\sigma_{\epsilon}^2$ (Canatar et al., 2021).

### A.2. Gaussian Universality Ansatz and the connection to Linear Random Feature Ridge Regression

As in (Simon et al., 2023; Atanasov et al., 2022), we adopt the Gaussian universality ansatz, which states that the expected train and test errors are unchanged if we replace $\{\zeta_t\}$ and $\{\phi_t\}$ with random Gaussian functions $\{\tilde{\zeta}_t\}$ and $\{\tilde{\phi}_t\}$ such that $\tilde{\zeta}_t(v) \sim \mathcal{N}(0, 1)$ and $\tilde{\phi}_t(x) \sim \mathcal{N}(0, 1)$ for $v \sim \mu_v$ and $x \sim \mu_x$, respectively.

The finite-feature stochastic kernels can then be written $\hat{H}^k(x, x') = \tilde{\theta}^k(x) \cdot \tilde{\theta}^k(x')$ where $\tilde{\theta}^k(x) \equiv Z^k \theta(x)$ and the entries of $Z^k \in \mathbb{R}^{N \times H}$ are drawn i.i.d. as $\mathcal{N}(0, 1/N)$ and we have defined $H$ to be the (possibly infinite) dimensionality of the reproducing kernel Hilbert space (RKHS) of $H$. The learned functions can then be written as

$$f^k(x) = \hat{w}^k \cdot \psi(x) \qquad , \qquad w^k = Z^{k\top} \left( Z^k \Theta^{\top} \Theta Z^{k\top} + \lambda I \right)^{-1} Z^k \Theta^{\top} y \tag{A.12}$$

Where $\Theta = [\theta(x_1), \cdots, \theta(x_P)]$, and the vectors $\theta(x_p) \sim \mathcal{N}(0, \Lambda)$, with $\Lambda$ a diagonal matrix with $\lambda_{tt} = \eta_t$. This is precisely the setting of a *linear* random-feature model with *data* covariance spectrum $\{\eta_1, \eta_2, ...\}$ (Atanasov et al., 2024). Under the Gaussian universality ansatz, we can therefore re-cast RFRR as linear RFRR with the role of the spectrum of the "data" played by the spectrum $\{\eta_t\}_{t=1}^{\infty}$ of the limiting deterministic kernel $H$.

## B. Notions of Deterministic Equivalent Error

Here, we review the notions of deterministic equivalent error relating the "true" error $\mathcal{E}_g^K$ of a RFRR ensemble to its deterministic approximation $E_g^K$. While $\mathcal{E}_g^K$ is a random quantity, depending on the particular realization of the dataset $\mathcal{D}$ as well as the random-feature weights $\{v_n^k\}$. However, many works have found that the stochastic quantity $\mathcal{E}_g^K$ *concentrates* about a *deterministic equivalent* error $E_g^K$ which depends on the dataset and random features only through their sizes $P$, $N$, and $K$ (see eq. 20). The exact relationship between $\mathcal{E}_g^K$ and $E_g^K$, however, varies in the literature. Here we describe two notions of deterministic equivalence, and discuss how and when they apply to the relationship $\mathcal{E}_g^K \simeq E_g^K$ employed in this work.

To derive a deterministic equivalent error formula, many works examine an "asymptotic" regime, in which both $N, P \to \infty$ with a fixed ratio ratio $P/N \sim \mathcal{O}(1)$. This permits the use of tools from random matrix theory to analyze the learned predictor. This approach is reviewed in (Atanasov et al., 2024), and we defer the reader to the citations therein. Under the

gaussian equivalence assumption which we employ, eq. 20 is a straightforward application of the error formula for linear random feature models (Atanasov et al., 2024; 2022; Adlam & Pennington, 2020; Maloney et al., 2022), with the role of the "data" played by the infinite-dimensional features of the RKHS of the limiting kernel $\boldsymbol{H}(\boldsymbol{x}, \boldsymbol{x}')$. While asymptotic equivalence holds only in the limit $N, P \to \infty$, the resulting formulas often serve as a good approximation at finite $N$ and $P$

A stronger notion of deterministic equivalence between $\mathcal{E}_g^1$ and $E_g^1$ was recently proven by Defilippis et al. *without* the assumption of Gaussian Equivalence, resolving the conjecture of (Simon et al., 2023) for single models ($K = 1$). Specifically, Defilippis et al. proved a "dimension-free" deterministic equivalence which guarantees that

$$|\mathcal{E}_g^1 - E_g^1| \leq C \left( \tfrac{1}{\sqrt{N}} + \tfrac{1}{\sqrt{P}} \right) E_g \,,$$

with high probability, for a factor $C$ which is logarithmically small in $N$ and $P$. Unlike asymptotic deterministic equivalence, dimension-free deterministic equivalence applies at fixed, finite $P$ and $N$. While at the time of writing the dimension-free deterministic equivalence has not been proven the the case of ensembled RFRR ($K > 1$), we anticipate that the arguments of (Defilippis et al., 2024) could be extended to the ensembled case.

## C. Proof of Theorems 4.1 and 5.1

In this section, we will refer to the condition that $\sum_t \bar{w}_t^2 \eta_t > 0$ as the task having a "learnable component." It can be shown from eq. 14 that $\mathrm{Df}_1 \leq \min(N, P)$. Because $N$ and $P$ are finite and $\mathrm{rank}(\{\eta_t\}_{t=1}^\infty)$ is infinite, it follows that $\kappa_2 > 0$. The following inequalities then hold strictly:

$$\mathrm{Df}_2 < \mathrm{Df}_1 \qquad\qquad \gamma_2 < \gamma_1 \qquad\qquad 0 < \rho < 1 \qquad\qquad \text{(C.1)}$$

We verify all deterministic equivalent error formulas used here by comparing to numerical experiments.

Furthermore, the inequality $\mathrm{tf}_2(\kappa) \leq \mathrm{tf}_1(\kappa)$ holds, with strict inequality when the target has a learnable component. In each of the proofs that follow, we aim to show that the error of a random-feature ensemble decreases monotonically under certain transformations of the parameters $(K, N, P) \to (K', N', P')$ when the value of the ridge parameter $\lambda$ is set to it's optimal value. We will argue using the invariance-based method introduced by (Simon et al., 2023). Denote the errors before and after the transformation by $E_g^K(N, P, \lambda)$ and $E_g^{K'}(N', P', \lambda')$. We allow here for the regularization $\lambda$ before the transformation to take an arbitrary value. Then, we will prove that there always exists a value $\lambda'$ such that $E_g^{K'}(N', P', \lambda') \leq E_g^K(N, P, \lambda)$. Proof follows by setting $\lambda$ equal to it's optimal value $\lambda^*$ and noting that $\min_\lambda E_g^{K'}(N', P, \lambda) \leq E_g^{K'}(N', P', \lambda') \leq E_g^K(N, P, \lambda^*)$. Below, we will prove the existence of $\lambda'$ under the desired transformations.

### C.1. "Bigger is Better" Theorems

We begin with the setting of the 4.1, so that we must prove error decreases under any transformation $(K, N, P) \to (K', N', P')$ such that $K \leq K'$, $N \leq N'$, and $P \leq P'$. We note that it suffices to prove that $E_g^K$ decreases monotonically with $K$, $N$, and $P$ separately, since any transformation $(K, N, P) \to (K', N', P')$ can be taken in steps $K \to K'$, $N \to N'$, $P \to P'$.

**Monotonicity with $K$**   The fact that when $K' > K$ and all other variables are held fixed, $E_g^K$ decreases is immediately evident from the form of eq. 20, because $\mathrm{Bias}_z^2$ and $\mathrm{Var}_z$ are independent of $K$. Furthermore, the inequality is strict as long as $\mathrm{Var}_z > 0$, which is valid as long as the task has a learnable component.

**Monotonicity with $P$**   Consider a transformation $P \to P'$ with $P' > P$. Examining eq. 14, we see that it is always possible to increase the ridge $\lambda$ such that $\kappa_2$ remains fixed. We then rewrite $E_g^K$ as:

$$E_g^K = (1 - \frac{1}{K}) \mathrm{Bias}_z^2 + \frac{1}{K} E_g^1 \qquad\qquad \text{(C.2)}$$

With $\kappa_2$ and $N$ fixed, we see that only the pre-factor of $\frac{1}{1-\gamma_2}$ in $E_g^1$ will be affected, so that $E_g^1$ decreases with $P$. Note also that $\gamma_1$ is a decreasing function of $P$. With $\kappa_2$ and $N$ fixed, it follows that $E_g^1$ decreases with $P$. Finally, because $K$ is fixed, $E_g^K$ will decrease with $P$.

**Monotonicity with** $N$    Consider a transformation $N \to N'$ with $N' > N$. Examining eq. 14, we see that it is always possible to increase the ridge $\lambda$ so that $\kappa_2$ remains constant. With $\kappa_2$, $P$ fixed, $\mathrm{Bias}_z^2$ is fixed as well. From eq. C.2, it then suffices to show that $E_g^1$ decreases. To see this, recall that $\rho$ is an increasing function of $N$, and $\gamma_1$ is a decreasing function of $\rho$. We then have that $\gamma_1$ is a decreasing function of $N$, so that the pre-factor of $\frac{1}{1-\gamma_1}$ in eq. 12 is decreasing with $N$.

### C.2. "No Free Lunch" from Ensembles Theorem

We now prove theorem 5.1. We first recall the form of the error to be:

$$E_g^K = \mathrm{Bias}_z^2 + \frac{1}{K}\left(E_g^1 - \mathrm{Bias}_z^2\right) \tag{C.3}$$

We define the variable $\nu \equiv \frac{1}{K}$. By analytical continuation, it suffices to show that test risk decreases as $\nu$ increases when $M = KN$ is held fixed. Rewriting the self-consistent equation in terms of $\nu$, we have;

$$\kappa_2 = \frac{\lambda N}{(P - \mathrm{Df}_1(\kappa_2))(\nu M - \mathrm{Df}_1(\kappa_2))} \tag{C.4}$$

Consider a transformation $\nu \to \nu'$ where $\nu' > \nu$. We se that it is always possible to increase $\lambda$ so that $\kappa_2$ remains fixed. Note that $\mathrm{Bias}_z^2$ depends only on $\kappa_2$ and $P$, so that as $\nu$ (and therefore $N$) vary, $\mathrm{Bias}_z^2$ remains fixed. From eq. 20, it therefore suffices to show that $\nu(E_g^1 - \mathrm{Bias}_z^2)$ decreases with $\nu$. Rearranging terms, we have:

$$\nu(E_g^1 - \mathrm{Bias}_z^2) = \nu\left[\frac{-\rho\kappa_2^2\,\mathrm{tf}_1'(\kappa_2) + (1-\rho)\kappa_2\,\mathrm{tf}_1(\kappa_2)}{1-\gamma_1} - \frac{-\kappa_2^2\,\mathrm{tf}_1'(\kappa_2)}{1-\gamma_2}\right] \tag{C.5}$$

$$+ \nu\left[\frac{1}{1-\gamma_1} - \frac{1}{1-\gamma_2}\right]\sigma_\epsilon^2 \tag{C.6}$$

We first show that

$$\frac{d}{d\nu}\left[\nu\left(\frac{1}{1-\gamma_1} - \frac{1}{1-\gamma_2}\right)\right] < 0, \tag{C.7}$$

To see this, recall that $\rho = (\nu M - \mathrm{Df}_1)/(\nu M - \mathrm{Df}_2)$. Because $\mathrm{Df}_1 > \mathrm{Df}_2$, this is a monotonically increasing function of $\nu$. We may write $\gamma_1 = \frac{1}{P}[(1-\rho)\mathrm{Df}_1 + \rho\,\mathrm{Df}_2]$. From this equation it is clear that $\gamma_1 > \gamma_2$. Differentiating with respect to $\nu$, we get

$$\frac{d\rho}{d\nu} = M\frac{(\mathrm{Df}_1 - \mathrm{Df}_2)}{(\nu M - \mathrm{Df}_2)^2} \tag{C.8}$$

$$\frac{d\gamma_2}{d\nu} = -\frac{1}{P}(\mathrm{Df}_1 - \mathrm{Df}_2)\frac{d\rho}{d\nu} = -\frac{M}{P}\frac{(\mathrm{Df}_1 - \mathrm{Df}_2)^2}{(\nu M - \mathrm{Df}_2)^2} \tag{C.9}$$

Using these, we have:

$$\frac{d}{d\nu}\left[\nu\left(\frac{1}{1-\gamma_1} - \frac{1}{1-\gamma_2}\right)\right] \tag{C.10}$$

$$= \left(\frac{1}{1-\gamma_1} - \frac{1}{1-\gamma_2}\right) + \frac{\nu\frac{d\gamma_1}{d\nu}}{(1-\gamma_1)^2} \tag{C.11}$$

$$= \frac{(\mathrm{Df}_1 - \mathrm{Df}_2)^2}{(1-\gamma_1)(\nu M - \mathrm{Df}_2)}\left[\frac{1}{P(1-\gamma_2)} - \frac{M\nu}{P(1-\gamma_1)(\nu M - \mathrm{Df}_2)}\right] \tag{C.12}$$

$$< 0 \tag{C.13}$$

where in the last line, we have used the facts that $\gamma_1 > \gamma_2$ and $\mathrm{Df}_2 \le \nu M$. To show that

$$\frac{d}{d\nu}\left[\nu\left(\frac{-\rho\kappa_2^2\,\mathrm{tf}_1'(\kappa_2) + (1-\rho)\kappa_2\,\mathrm{tf}_1(\kappa_2)}{1-\gamma_1} - \frac{-\kappa_2^2\,\mathrm{tf}_1'(\kappa_2)}{1-\gamma_2}\right)\right] \le 0, \tag{C.14}$$

we first note that $-\rho\kappa_2^2\,\mathrm{tf}_1'(\kappa_2) + (1-\rho)\kappa_2\,\mathrm{tf}_1(\kappa_2)$ can be equivalently written as $-\kappa_2^2\,\mathrm{tf}_1'(\kappa_2) + (1-\rho)\kappa_2\,\mathrm{tf}_2(\kappa_2)$. The above derivative can then be broken into two parts:

$$-\kappa_2^2\,\mathrm{tf}_1'\,\frac{d}{d\nu}\left[\nu\left(\frac{1}{1-\gamma_1} - \frac{1}{1-\gamma_2}\right)\right] + \kappa_2\,\mathrm{tf}_2\,\frac{d}{d\nu}\left[\frac{\nu(1-\rho)}{1-\gamma_1}\right] \tag{C.15}$$

We have already shown that the derivative in the first term is negative. Furthermore, $-\kappa_2^2\,\mathrm{tf}_1' \ge 0$, with strict equality holding when the task has a learnable component. To see that the derivative in the second term is negative, note that $\rho$ is an increasing function of $\nu$. Because $\gamma_1$ is a decreasing function of $\rho$, $\gamma_1$ is therefore a decreasing function of $\nu$. The denominator $1-\gamma_1$ inside the derivative increases with $\nu$. Furthermore, the numerator $\nu(1-\rho)$ can be written as $(\mathrm{Df}_1 - \mathrm{Df}_2)\frac{\nu}{\nu M - \mathrm{Df}_2}$. With $\kappa_2$ fixed (so that $\mathrm{Df}_1 - \mathrm{Df}_2 > 0$ is fixed), this is a strictly decreasing function of $\nu$. It follows that

$$\kappa_2\,\mathrm{tf}_2\,\frac{d}{d\nu}\left[\frac{\nu(1-\rho)}{1-\gamma_1}\right] \le 0, \tag{C.16}$$

with strict inequality holding as long as $\mathrm{tf}_2 > 0$, which is true whenever the task has a learnable component.

## D. Derivation of Scaling Laws

In this section, we derive the width-bottlenecked scaling laws given in section 6.2, using methods described in (Atanasov et al., 2024). We assume that the kernel eigenspectrum decays as $\eta_t \sim t^{-\alpha}$ and the power of the target function in the modes decays as $\bar{w}_t^2\eta_t \sim t^{-(1+2\alpha r)}$, and examine the regime where $P \gg N$, so that the width of the ensemble members is the bottleneck to signal recovery. We begin by analyzing the self-consistent equation for $\kappa_2$, reproduced here for clarity:

$$\kappa_2 = \frac{\lambda N}{(P - \mathrm{Df}_1(\kappa_2)(N - \mathrm{Df}_1(\kappa_2)))} \tag{D.1}$$

Because $\mathrm{Df}_1(\kappa_2) < \min(N, P)$ and $N \ll P$, it follows that $P \gg \mathrm{Df}_1(\kappa_2)$. We can therefore approximate the fixed-point equation as

$$P\kappa_2 \approx \frac{\lambda}{1 - \frac{1}{N}\mathrm{Df}_1(\kappa_2))} \tag{D.2}$$

We approximate $\mathrm{Df}_1$ using an integral:

$$\mathrm{Df}_1(\kappa_2) \approx \int_1^\infty \frac{t^{-\alpha}}{t^{-\alpha} + \kappa_2}dt \tag{D.3}$$

Making the change of variables $u = t\kappa_2^{1/\alpha}$, we get

$$\mathrm{Df}_1(\kappa_2) \approx \kappa_2^{-1/\alpha}\int_{\kappa_2^{1/\alpha}}^\infty \frac{du}{1+u^\alpha} \tag{D.4}$$

Plugging back into the fixed-point equation, we arrive at

$$\kappa_2 P \approx \frac{\lambda}{1 - \frac{\kappa_2^{-1/\alpha}}{N}\int_{\kappa_2^{1/\alpha}}^\infty \frac{du}{1+u^\alpha}} \tag{D.5}$$

Next, we make the ansatz that $\kappa_2 \sim N^{-q}$. The fixed point equation becomes

$$PN^{-q} \sim \frac{\lambda}{1 - N^{\frac{q}{\alpha}-1}\int_{N^{-q/\alpha}}^\infty \frac{du}{1+u^\alpha}} \tag{D.6}$$

The left size of this equation will be very large, due to the separation of scales $P \gg N$. The only feasible way for the right side to scale with $P$ is for the denominator to become very small as $N$ grows. This is only possible if $N^{q/\alpha} \sim N$, so that $q = \alpha$. We therefore have $\kappa_2 \sim N^{-\alpha}$.

With this scaling for $\kappa_2$, we have $\mathrm{Df}_1, \mathrm{Df}_2 \sim N$. It is then clear that $\gamma_2 \to 0$ for $P \gg N$. Furthermore, because $\rho \in [0, 1]$, $\gamma_1 \to 0$ for $P \gg N$. The pre-factors of $\frac{1}{1-\gamma_2}$ and $\frac{1}{1-\gamma_1}$ can therefore be ignored.

We may then write

$$\kappa_2 \, \mathrm{tf}_1(\kappa_2) \sim \int_1^\infty \frac{t^{-(1+2\alpha r)}}{1 + t^{-\alpha}/\kappa_2} dt \sim N^{-2\alpha r} \int_{1/N}^\infty \frac{u^{-(1+2\alpha r)}}{1 + u^{-\alpha}} du \tag{D.7}$$

where $u = t\kappa^{1/\alpha}$ and we have made the substitution $\kappa \sim N^{-\alpha}$. We get two contributions to the integral: when $u$ is near $1/N$, we get a contribution (including the pre-factor) which scales as $N^{-\alpha}$. When $u$ is away from $1/N$, the integral contributes a constant factor and we get a contribution that scales as the pre-factor $N^{-2\alpha r}$.

Similarly, we may write:

$$-\kappa_2^2 \, \mathrm{tf}_1'(\kappa_2) \sim \int_1^\infty \frac{t^{-(1+2\alpha r)}}{(1 + t^{-\alpha}/\kappa_2)^2} dt \sim N^{-2\alpha r} \int_{1/N}^\infty \frac{u^{-(1+2\alpha r)}}{(1 + u^{-\alpha})^2} du \tag{D.8}$$

The contributions from the component of the integral near $1/N$ will now scale as $N^{-2\alpha}$, and the contribution away from $1/N$ will remain $N^{-2\alpha r}$. Combining these results, we arrive at separate scaling laws for the bias and variance terms of the error:

$$\mathrm{Bias}_z^2 \sim N^{-2\alpha \min(r,1)} \tag{D.9}$$

$$\mathrm{Var}_z \sim N^{-2\alpha \min(r,\frac{1}{2})} \tag{D.10}$$

Finally, to obtain eq. 26, we put $N \sim M^\ell$ and $K \sim M^{1-\ell}$ and substitute into eq. 20. We find that, in terms of $M$, the bias and variance scale as:

$$\mathrm{Bias}_z^2 \sim M^{-2\ell\alpha \min(r,1)} \tag{D.11}$$

$$\frac{1}{K} \mathrm{Var}_z \sim M^{-\left(1-\ell+2\alpha\ell \min\left(r,\frac{1}{2}\right)\right)} \tag{D.12}$$

The scaling of the total loss for an ensemble will be dominated by the more slowly-decaying of these two terms.

### D.1. Sample-Bottlenecked Scaling

We next examine the case where $P \ll N$. Here, we find that $\mathrm{Df}_1(\kappa_2), \mathrm{Df}_2(\kappa_2) \ll N$ so that $\rho \to 1$, $\mathrm{Var}_z \to 0$, so that the most significant contribution to the error comes from the bias: The only significant contribution to the error will come from $\mathrm{Bias}_z^2$, which will scale as (Atanasov et al., 2024):

$$E_g^K \sim P^{-2\alpha \min(r,1)} \qquad\qquad (\lambda \ll P^{1-\alpha}) \tag{D.13}$$

$$E_g^K \sim (\lambda/P)^{2\min(r,1)} \qquad\qquad (\lambda \gg P^{1-\alpha}) \tag{D.14}$$

We therefore see that the ensemble size $K$ and network size $N$ have no effect on the scaling law in $P$ provided $P \ll N$.

### D.2. Overparameterized Ensembles with Small Ridge

Here, we analyze the behavior of ensembles of RFRR models in the overparameterized regime ($N \gg P$) and with small ridge $\lambda$. We set the scale of the noise $\sigma_\epsilon = 0$ for simplicity. We write the "renormalized ridge" $\kappa_2$ and degrees of freedom as

power series in the small parameters $1/N$ and $\lambda$, which we assume to be on the same order of magnitude.

$$\kappa_2 \approx \kappa_2^* + \delta/N + \alpha\lambda \tag{D.15}$$

$$\mathrm{Df}_1(\kappa_2) \approx P + \mathrm{Df}_1'(\kappa_2^*)(\delta/N + \alpha\lambda) \tag{D.16}$$

$$\mathrm{Df}_2(\kappa_2) \approx \mathrm{Df}_2(\kappa_2^*) + \mathrm{Df}_2'(\kappa_2^*)(\delta/N + \alpha\lambda), \tag{D.17}$$

for some parameters $\delta, \alpha$ which are $\mathcal{O}(1)$ in $1/N$ and $\lambda$. Plugging these into the fixed-point equation (eq. 14) and expanding about $\lambda = 1/N = 0$, we can solve for $\alpha, \delta$ by matching coefficients at first order:

$$\alpha = -\frac{1}{\kappa_2^* \,\mathrm{Df}_1'(\kappa_2^*)} \tag{D.18}$$

$$\delta = \frac{\lambda P \kappa_2^*}{2\lambda - \kappa_2^{*2}\,\mathrm{Df}_1'(\kappa_2^*)} \tag{D.19}$$

Finally, expanding equations 19 in power series in $1/N$ and $\lambda$, we obtain the following (for $\sigma_\epsilon = 0$):

$$\mathrm{Bias}_z^2 = -\frac{P \kappa_2^{*2}\,\mathrm{tf}_1'(\kappa_2^*)}{P - \mathrm{Df}_2(\kappa_2^*)} + \lambda F(\kappa_2^*, P) + \mathcal{O}(\lambda^2, \lambda/N, 1/N^2) \tag{D.20}$$

$$\mathrm{Var}_z = \frac{P \kappa_2^*\,\mathrm{tf}_1(\kappa_2^*)}{N} + \mathcal{O}(\lambda^2, \lambda/N, 1/N^2), \tag{D.21}$$

where

$$F(\kappa_2^*, P) \equiv \frac{P\left((P - \mathrm{Df}_2(\kappa_2^*))\,\mathrm{tf}_2'(\kappa_2^*) + \kappa_2^*\,\mathrm{Df}_2'(\kappa_2^*)\,\mathrm{tf}_1'(\kappa_2^*)\right)}{\mathrm{Df}_1'(\kappa_2^*)(P - \mathrm{Df}_2(\kappa_2^*))^2} \tag{D.22}$$

This gives us that, to leading order in $\lambda$ and $1/N$, the error of an ensemble of $K$ RFRR models, each with $N$ features, can be written as

$$E_g^K = -\frac{P \kappa_2^{*2}\,\mathrm{tf}_1'(\kappa_2^*)}{P - \mathrm{Df}_2(\kappa_2^*)} + \lambda F(\kappa_2^*, P) + \frac{P \kappa_2^*\,\mathrm{tf}_1(\kappa_2^*)}{KN} + \mathcal{O}(\lambda^2, \lambda/N, 1/N^2) \tag{D.23}$$

We note that, at leading order, this depends on ensemble size $K$ and model size $N$ only through the total number of features $KN$.

## E. RFRR on Real Datasets

### E.1. Numerical Experiments with Synthetic Gaussian Data

For a given value of $\alpha$ and $r$, we fix a large value $D \gg P, M$ and generate eigenvalues $\eta_t \propto t^{-\alpha}$ and target weights $\bar{w}_t \sim t^{-\frac{1}{2}(1-\alpha+2\alpha r)}$. The $\eta_t$ and $\bar{w}_t$ are normalized so that $\sum_t \eta_t = 1$ and $\sum_t \bar{w}_t^2 \eta_t = 1$. We generate random features as $\boldsymbol{\Theta} = \boldsymbol{\Sigma}\boldsymbol{X} \in \mathbb{R}^{D \times P}$, where $\boldsymbol{X}_{ij} \sim \mathcal{N}(0,1)$ and $\boldsymbol{\Sigma}$ is the diagonal matrix with entry $\boldsymbol{\Sigma}_{tt} = \eta_t$. Labels are assigned as $\boldsymbol{y} = \boldsymbol{\Psi}^\top \bar{\boldsymbol{w}} \in \mathbb{R}^P$. For each $k = 1, \ldots, K$, we perform linear RFRR (eq. A.12) with an independently drawn projection matrix $\boldsymbol{Z}^k$ with entries drawn from $\mathcal{N}(0, 1/N)$. The prediction of the ensemble is then given as the mean over the $K$ learned predictors. We use this procedure to generate numerical error curves in Figures 3 and S2.A-C.

### E.2. Numerical Experiments on Binarized MNIST and CIFAR-10 with ReLU Features

We perform RFRR on CIFAR-10 and MNIST datasets. To construct the dataset, we sort the images into two classes. For CIFAR-10, we assign a label $y = +1$ to images of "things one could ride" together (airplane, automobile, horse, ship, truck) and a label $y = -1$ for "things one ought not to ride" (bird, cat, deer, dog, frog) (Simon et al., 2023). For MNIST, we assign a label $y = +1$ to digits $0 - 4$, and a label $-1$ to digits $5 - 9$. We construct $K$ feature maps as $\psi(^k\boldsymbol{x}) = \frac{1}{\sqrt{N}} \mathrm{ReLU}\left(\boldsymbol{V}^{k\top}\boldsymbol{x}\right)$, where $\boldsymbol{V}_{ij}^k \sim \mathcal{N}(0, 2/D)$. Here, $D$ is the data dimensionality ($D = 3072$ for CIFAR-10 and 784 for MNIST). Then, for each ensemble member $k = 1, \ldots, K$, we train a linear regression model on the features $\psi^k$. In the infinite-feature limit, the finite-feature kernels will converge to the "NNGP kernel" for a single-hidden-layer Relu network (Lee et al., 2018). This procedure was used to generate numerical error curves for figures 1, 2, 4, S1, S2.D-F, S3, S4, S5, S6, and S7

### E.3. Theoretical Predictions

We evaluate the omniscient risk estimate 20 numerically using vectors storing the values of $\{\eta_t\}_{t=1}^{\infty}$ and $\{\bar{w}_t\}_{t=1}^{\infty}$. In the case of synthetic data, these vectors are readily available. For the MNIST and CIFAR-10 tasks, we approximate these vectors by evaluating the infinite-width Neural Network Gaussian Process (NNGP) kernel using the neural tangents library (Novak et al., 2019). For 30,000 images from the training sets of both MNIST and CIFAR 10, we evaluate the kernel matrix $[\boldsymbol{H}]_{p,p'} = \boldsymbol{H}(\boldsymbol{x}_p, \boldsymbol{x}_{p'})$, and diagonalize the kernel matrix to determine the eigenvectors and eigenvalues. To be precise, with $P = 30,000$, we calculate the eigenvalues $\{\eta_1, \eta_2, \ldots, \eta_P\}$ and eigenvectors $\{\boldsymbol{u}_1, \ldots, \boldsymbol{u}_P\}$ of the sample-normalized kernel matrix $\frac{1}{P}\boldsymbol{H}$. We then assign the weights of the target function as $\bar{w}_t = \frac{1}{\sqrt{P\eta_t}}\boldsymbol{u}_t^\top \boldsymbol{y}$, where $\boldsymbol{y} \in \mathbb{R}^P$ is the vector of labels associated to our $P$ samples.

We then solved the self-consistent equation (eq. 14) using the Bisection method of the scipy library (Virtanen et al., 2020), and evaluate eq. 20 to determine the predicted risk.

### E.4. Measuring Power Law Exponents of Kernel Ridge Regression Tasks

In fig. 4B, we plot theoretical predictions for the scaling exponents of the bias and variance contributions to error for the binarized CIFAR-10 and MNIST classification tasks. To calculate these exponents, we need access to the "ground truth" source and capacity exponents $\hat{\alpha}$ and $\hat{r}$ characterizing the kernel eigenstructure of the dataset and the target function, such that the eigenvalues $\{\eta_1, \eta_2, \ldots\}$ of the NNGP kernel decay as $\eta_t \sim t^{-\hat{\alpha}}$ and the weights of the target function $\{\bar{w}_1, \bar{w}_2, \ldots\}$ decay as $\eta_t \bar{w}_t^2 \sim t^{-(1+2\hat{\alpha}\hat{r})}$. To estimate $\hat{\alpha}$, we calculate the "trace metric" $\left[\text{tr}\left[\boldsymbol{H}_p^{-1}\right]\right]^{-1}$ $\hat{\alpha}$ is then obtained by fitting to the relationship $\left[\text{tr}\left[\boldsymbol{H}_p^{-1}\right]\right]^{-1} \sim p^{-\alpha}$, where $\boldsymbol{H}_p \in \mathbb{R}^{p \times p}$ is the empirical NNGP kernel for $p$ randomly drawn samples from the dataset (Wei et al., 2022; Simon et al., 2023) (figs. S7.A,C). To estimate the source exponent $r$, we use the scaling law for Kernel Ridge Regression (with the limiting infinite-feature NNGP kernel) which dictates that for small ridge, $E_g \sim p^{-2\alpha \min(r,1)}$ (see Appendix D.1). Following Simon et al., we fit the MSE loss for Kernel Ridge Regression with the limiting NNGP kernel to a power law decay $E_g \sim p^{-\beta}$ (figs. S7.B,D), and assign $\hat{r} = \beta/2/\hat{\alpha}$.

## F. Code Availability

All code used to generate the figures presented in this work is publicly available at https://github.com/benruben87/Random_Feature_Ensembles.git.

