# OpenReview forum: "No Free Lunch from Random Feature Ensembles: Scaling Laws and Near-Optimality Conditions"
_ICML.cc/2025/Conference — ICML 2025 poster_

### Official Review · Reviewer_5sWk · 2025-03-06

**Overall Recommendation:** 3

**Summary:**

The paper investigates the random-feature ridge regression between using a single large model versus multiple smaller models (ensembles). The authors demonstrate that ensembles can achieve near-optimal performance when the total feature count remains high in the overparameterized regime, while in the underparameterized regime, error scaling laws depend on the relationship between ensemble size and model size.

**Claims And Evidence:**

The claims are supported by clear evidence.

**Essential References Not Discussed:**

I have some concerns on the essential contribution of this paper. The rigorous excess risk curve has already been investigated by previous papers.  [1] gave the rigorous theoretical values of risks of random feature model,  and [2] extended the single random feature model to ensembled random feature model. With $y=f_0(x)+\epsilon$, people can also easily get rigorous theoretical values of risks of ensembled random feature model based on the analysis of [1] and [2]. The authors should discuss more on these two references.


[1] Mei S, Montanari A. The generalization error of random features regression: Precise asymptotics and the double descent curve[J]. Communications on Pure and Applied Mathematics, 2022, 75(4): 667-766.

[2] Meng, X., Yao, J., & Cao, Y. (2024). Multiple descent in the multiple random feature model. Journal of Machine Learning Research, 25(44), 1-49.

**Experimental Designs Or Analyses:**

In Section D.1, D.2, the authors should clearly point out where are the experimental results.

**Methods And Evaluation Criteria:**

I appreciate the real data experiments via CIFAR10 and MNIST. The authors  performed fully numerical experiments first to support their theory first. Then they also conducted real data experiments.

**Other Comments Or Suggestions:**

See above.

**Other Strengths And Weaknesses:**

The paper is generally well written. However, there are two main concerns.

[1] Literature related issues given above. Please also see my questions below.
[2] Clearer resources in Section 3.1.

**Questions For Authors:**

Can the authors rely on the results presented in [1] and [2]? In my understanding, the solution for random feature models in [2] directly corresponds to the solution of ensembled random feature models. This follows from the fact that for different vectors  $x_1, x_2$,


$\min_{x_1, x_2} \left( f(x_1) + g(x_2) \right) = \min_{x_1} f(x_1) + \min_{x_2} g(x_2)$,


which implies that


$x_1^* = \arg\min f(x_1), \quad x_2^* = \arg\min g(x_2)$.

**Relation To Broader Scientific Literature:**

The authors missed two important references, which already rigorously gave the risks of single random feature model and ensembled random feature models. The authors should discuss these references adequately, and compare the results.

[1] Mei S, Montanari A. The generalization error of random features regression: Precise asymptotics and the double descent curve[J]. Communications on Pure and Applied Mathematics, 2022, 75(4): 667-766.

[2] Meng, X., Yao, J., & Cao, Y. (2024). Multiple descent in the multiple random feature model. Journal of Machine Learning Research, 25(44), 1-49.

**Theoretical Claims:**

I did not check how equation (9) comes. To my understanding, all the comparison below comes from equation (9), the authors should give clearer demonstration on Section 3.1, at least give clear resources where these equations come from.

---

> ### Author Rebuttal · Authors · 2025-04-01
>
> Thank you for your review.  We respond to your questions concerns as follows:
>
> > In Section D.1, D.2, the authors should clearly point out where are the experimental results.
>
> **Response:** Thank you for your suggestion.  We will update the text to point this out.
>
> Section D.1 on synthetic tasks pertains to the numerical error points in Figures 3 and S2.A-C.
>
> Section D.2 on the numerical experiments for  binarized MNIST and CIFAR-10 with ReLU features pertains to figures 1, 2, 4, S1, S2.D-F, S3, S4, S5, S6, and S7.
>
>
> > The authors missed two important references ([1], [2])...compare the results.
>
> Thank you for bringing these works to our attention.  We were aware of ref [1] and many other papers deriving the error... These papers focus on the over-fitting and double descent / multiple descent pheonomena in their analysis.  However, these effects vanish entirely when the ridge parameter is optimized at each sample size.  By studyiing the behavior of the test risk at optimal ridge, we obtain more practical results that will apply even in settings where the ridge parameter is set to its optimal value using cross-validation.
>
> > I have some concerns on the essential contribution ... these two references.
> > Can the authors rely on the results ...$ \arg\min_x f_2(x).
> $
>
> Thank you for bringing these papers to our attention.  In short, no, we cannot rely on either of them to derive the test risk of ensembled RFRR.
>
> As for ref. [1], the data model considered here consists of samples drawn uniformly from the unit sphere, whereas the results we use apply to arbitrary (but nicely behaved) data distributions $\mathbf{x} \sim \mu_{\mathbf{x}}$.
>
> As for ref. [2], the loss function considered there encodes \textit{joint} training of the parameters $\mathbf{a}\_1 \equiv [a\_1, \dots, a\_{N\_1}]^\top $ and $\mathbf{a}\_2 \equiv [a\_{N\_1 + 1}, \dots, a\_{N\_2}]^\top$.  This can be written as
>
> $$ \frac{1}{n} \sum_{i=1}^n \left( y - f_1(\mathbf{a}\_1) \right)^2 + \frac{d}{n} \lambda ||\mathbf{a}\_1||^2  + \frac{1}{n} \sum_{i=1}^n \left( y - f_1( \mathbf{a}\_1) \right)^2 + \frac{d}{n} \lambda  ||\mathbf{a}\_2||^2 $$
>
> The square term here cannot be factored into a sum of contributions depending separately on $\mathbf{a}\_1$ and $\mathbf{a}\_2$, as you have assumed in your comment.
>
> In the notation of [2], the model we consider corresponds to a decoupled loss function which optimizes $\mathbb{a}\_1$ and $\mathbf{a}\_2$ separately:
>
> $$\frac{1}{n} \sum_{i=1}^n\left( y - f_1(\mathbf{a}\_1) \right)^2 + \frac{d}{n} \lambda ||\mathbf{a}\_1||^2+ \frac{1}{n} \sum\_{i=1}^n\left( y - f_1(\mathbf{a}\_1) \right)^2 + \frac{d}{n} \lambda  ||\mathbf{a}\_2||^2 \,,$$
>
> with additional terms if $K>2$.
>
>
> > Additional response to reviewer comments:
>
> Many of your questions and concerns revolve around the derivation of the risk estimate for random feature ensembles, and its presence in prior literature.  There are many papers which derive the bias and variance terms for random-features regression using a variety of methods, which we have referenced (Atanasov e tal.,2024; Canatar et al. ,2021; Simon et al., 2023; Adlam & Pennington, 2020; Rocks & Mehta, 2021; Hastie etal., 2022; Zavatone-Vethetal., 2022) and we are happy to add discussions of [1] and [2].
>
> We do not claim to have contributed the error formula for ensembles -- our contribution is a novel and informative analysis of test risk formula.
>
> The error formulas for RFRR ensembles derived in previous works are not easily interpretable, and studying the implications of these error formulas is an important task unto itself.  We believe that our paper has addressed an important gap in our understanding of ensembled random feature models. We have used the (known) bias-variance decomposition of the test risk of RFRR to study the tradeoff between model size and ensemble size.  Specifically, we have shown that:
> - Ensembling is *never* optimal under a fixed parameter budget at optimal ridge.
> - Ensembling can achieve near-optimal performance in both the overparameterized and underparameterized regimes, with precise spectral conditions for near-optimal scaling in the underparameterized regime.
> If accepted, we will clarify these contributions in the abstract and introduction.  We may also change the title of the paper to "No Free Lunch from Random Feature Ensembles: Scaling Laws and Near-Optimality Conditions" to highlight all of our main contributions.
>
>
> [1] Mei S, Montanari A. The generalization error of random features regression: Precise asymptotics and the double descent curve J. Communications on Pure and Applied Mathematics, 2022, 75(4): 667-766.
>
> [2] Meng, X., Yao, J., & Cao, Y. (2024). Multiple descent in the multiple random feature model. Journal of Machine Learning Research, 25(44), 1-49.

---

### Official Review · Reviewer_Ef22 · 2025-03-11

**Overall Recommendation:** 4

**Summary:**

## Updates after author discussion

Thanks a lot for all the clear discussion. A lot of my issues/confusion with the paper have been addressed in the comments, and I'm convinced that the theory just needs some cleaning up to be fully clear. The paper then tells an interesting -- and, to my knowledge, novel -- story about ensembling. I've increased my score to a 4/5 to vote for an accept. I would just recommend to the authors to do a few careful scrubbing passes over the theoretical sections to make sure notation is being used consistently and all the asymptotics are carefully defined *and* explained (e.g., the fact that $\lambda \to 0$ is (1) done for analytical convenience, but (2) has some justification in the literature is helpful discussion that should definitely be in the paper!)

Best,
Reviewer Ef22


## Original review before author response
This is a theoretical paper studying kernel ridge regression with random features. Specifically, it asks whether ensembles of multiple models are effective when the optimal ridge parameter is used. The authors show that, in fact, when holding the total number of random features constant across all models (representing a fixed compute budget), the test risk is minimized by not ensembling. However, the authors note that ensembles can achieve *nearly* optimal performance in the overparameterized regime. The authors validate their theory in small-scale synthetic and real experiments, which point to some interesting future directions around when the ridge parameter is not optimally chosen.

**Claims And Evidence:**

I think that the theoretical results, as stated, back up the overall premise of the paper. And the empirical results also effectively illustrate the theoretical results. I do have some issues with the theoretical development, which are listed below.

The one portion of the paper that I didn't see as offering much evidence was Section 6. This felt a little tacked on, and I wasn't sure how it related to the main story of the paper, which is (to my reading) about ensembling not being effective. I think some more discussion of what the motivation / takeaways of this section are would be helpful.

**Essential References Not Discussed:**

Nothing as far as I know!

**Experimental Designs Or Analyses:**

Yes, the paper is mostly theoretical, so there isn't too much choice of experimental design here.

**Methods And Evaluation Criteria:**

Yes, the paper is mostly theoretical, so there isn't too much choice of methodology here.

The only comments I had were from reading Appendix D (details on experimental setups):
1. Section D.1 states that that the $\eta_t$ are normalized so that $\sum_t \eta_t = 1$ and $\sum_t \bar w_t^2 \eta_t = 1$. Why is it that these are both simultaneously satisfiable?
2. In Section D.2, why is the variance of $V_{ij}^k$ chosen to be $2/D$? Is this a result from Lee et al. (2018)? If so, some quick rewording might clarify this.

**Other Comments Or Suggestions:**

I would just point out that, while there's not a statistical gain in ensembling here (in fact it seems there may be a statistical loss!) there is a computational gain, as the compute required for $M$ features is $O(M^3)$ versus $M$ features spread over $K$ ensembles requires $O(M^3 / K^2)$. I think this would be good to bring up around the discussion of Theorem 5.1.

**Other Strengths And Weaknesses:**

I've listed everything in the sections above.

**Questions For Authors:**

My main questions that influenced my review are:

1. Are all the approximate equalitites really justified in the proofs / theoretical statements?
2. Can the proof of theorem 5.1 be expanded on or my understanding of it corrected?
3. What is the purpose of Section 6?

**Relation To Broader Scientific Literature:**

Ensembling is an important idea in the machine learning literature; e.g. random forests are hugely successful in practice, and ensembles of deep models have been proposed recently for various purposes such as uncertainty estimation. As far as I know, most work on ensembling does not study the very important tradeoff in terms of number of parameters (representing compute) and statistical accuracy. I think this paper is an interesting step towards understanding that tradeoff.

**Theoretical Claims:**

There were a few times where I thought the theoretical development was hard to follow because it seemed to be lacking definition, was not fully rigorous, or seemed to have skipped a few steps in proofs. I've bulleted out these issues below:

**Lacking definition**
1. "consider the "featurization" transformation $g$" -- I think this could really use a precise example to clarify what $g$ is supposed to be.
2. $\mu_v$ was used around the second column of line 62, but I didn't see a definition for it.
3. ``As $N \to \infty$, this stochastic kernel converges to the deterministic kernel $H(x,x')$''. Does this not require a careful choice of $g$ and $\mu_v$.
4. ``$f(x)$ and the true target function $f(x)$'' -- I think this is just a typo
5. "where $\mathcal{E}_g^1$ is the "true'' risk" -- How is the definition of $E^1_g$ in equation 9 not the `"true" risk? I didn't really understand what $\mathcal{E}$ is supposed to be.
6. Eq 7 seems to define $E_g$ for a fixed $f$. But then Eqs 12-14 decompose $E_g$ using expectations over $Z$ (i.e., over the random features making up $f$). These seem inconsistent with one another.
7. In section 5.1, "$\lambda = 1/N = 0$". It doesn't seem possible that $\lambda$ could be both $1/N$ and 0.

**Missing Rigor**

1. I think it's fine to not show the derivation of Eq 9. But it's written as an approximate equality without ever saying what the slack in this approximation is. I think this should be stated.
2. Eq 21 seems like a major point of the paper, given that it's the main result showing near optimality of ensembling in the overparameterized regime. But, it's proof in Appendix C.2 doesn't seem fully rigorous. First, in the proof $1/N$ and $\lambda$ are stated to be "[assumed] to be on the same order of magnitude" without stating what this means. Second, the proof uses a number of approximate equalities (Eqs C.15-C.17) without specifying what the approximation is or how it affects the proof. Overall, I think this result should be wrapped into a lemma / theorem environment with carefully stated assumptions and a complete proof.
3. In Appendix C (derivation of the scaling laws), there were a lot of approximations used. It's not clear how these affect the proof.

Overall, my major issue with rigor is that I don't think it's appropriate to use approximate equalities in a formal proof without precisely defining what terms $\approx$ sign is hiding and verifying that dropping those terms doesn't affect the result.

**Missing Steps in Proofs**

Appendix B.1 contains the proof of Theorem 4.1, which says that when minimizing the test risk over the ridge parameter $\lambda$, the test risk decreases monotonically with the number of features, number of ensemble elements, and number of training datapoints. It's overall not clear to me how this proof accounts for the fact that we are minimizing over $\lambda$. There's a reference to Eq 11 and the fact that its fixed point can be held constant by scaling up $\lambda$. But we're optimizing over $\lambda$. So I'm not totally sure what this is pointing out.

Also, the proof of monotonicity in $N$ (the number of model features) uses Eq. 9 to derive its result. But Eq 9 is an approximate equality, so I don't think that analyzing it can lead us to rigorous conclusions.

---

> ### Author Rebuttal · Authors · 2025-04-01
>
> Thank you for your detailed review. Below, we address your questions and concerns:
>
> > "The one portion ... would be helpful."
>
>  **Response**: Thank you for this suggestion. We will add more justification. While ensembles are never optimal, they allow parallelization and can be *near-optimal*, hence useful in practice. Section 5.1 addresses near-optimality in the overparameterized regime.  The scaling laws derived in section 6 address near-optimality in the under-parameterized regime -- necessary for a complete characterization.
>
> > 1. **Section D.1** states ... satisfiable?
>
> **Response:** We will clarify that both the $\eta_t$ and $\bar{w}_t$ are normalized to satisfy these constraints.
>
> > 2. In **Section D.2**, why ... clarify this.
>
> **Response:** Yes, we choose $\mathbf{V}^k_{ij}\sim \mathcal{N}(0, 2/D)$ to converge to the NNGP kernel in Lee et. al. (2018).  We will clarify this in the text.
>
> >1. **"consider the *featurization* transformation $g$"**. I ... supposed to be.
> >2. **$\mu_\nu$** was ... for it.
>
> **Response**: We have revised the text:
>
> "Define the random features $\mathbf{\psi}(\mathbf{x}) \in \mathbb{R}^N$ by $\left[ \mathbf{\psi}(\mathbf{x})\right]\_n = g(\mathbf{v}\_n, \mathbf{x})$, where the $\mathbf{v}\_n$ are random parameter vectors sampled independently from measure $ \mu\_{\mathbf{v}}$ on $\mathbb{R}^C$.  Here, the function  $g: \mathbb{R}^C \times R^{D} \mapsto \mathbb{R}$ is a "featurization transformation," often taking the form $g(\mathbf{v}\_n, \mathbf{x}) = \varphi (\mathbf{v\_n}^\top \mathbf{x})$ for some nonlinear activation function $\varphi(\cdot)$... "
>
> >3. **"As $N \to \infty$, this... and $\mu_\nu$?
>
> **Response**:  A careful choice of $g$ and $\mu\_{\mathbf{v}}$ is required to ensure convergence to *a particular* kernel (see point 2).  However, the stochastic kernel in general converges to *some* deterministic kernel given by $H(x, x') = \mathbb{E}\_{\mathbf{v} \sim \mu\_{\mathbf{v}}} g(\mathbf{v}, \mathbf{x})g(\mathbf{v}, \mathbf{x'})$.  We will clarify.
>
> >6. **Equation (7)** seems to ... with one another.
>
>    **Response**: This is a standard bias-variance decomposition of the error (see, [2]).  An expectation over $\mathbf{Z}$ is not necessary on the left side of eq. 12 because the test risk **concentrates** over $\mathbf{Z}$.
>
> >7. In **Section 5.1**, *"$\lambda = 1/N = 0$"*.... $1/N$ and 0.
>
>    **Response**: To clarify, we now say "we expand the risk estimate $E_g^K$ (eq. 9) as power series in $1/N \gtrsim 0$ and $\lambda  \approx 0$".
>
> > Comments regarding the use of eq. 9, and the slack in this estimate ("where $\mathcal{E}\_g^1$ is ... to be."; "I think... should be stated."; "In Appendix C ... result."; "Also, the proof rigorous conclusions."):
>
> **Response**: Thank you, we agree that more thorough explanation would be helpful. We call the "true" risk $\mathcal{E}\_g$ the error for a particular realization of the random parameters $\mathbf{v}\_n$.  The key result of Defilippis *et al.* (2024) is to show that the *distribution* over $\mathcal{E}\_g$ values *concentrates* around a *deterministic‐equivalent* expression $E\_g$ which depends on the feature count $N$. The difference between the *true* random‐features generalization error $\mathcal{E}\_g$ and its deterministic‐equivalent **$E\_g$** is controlled by a **multiplicative concentration bound**:
> $$ |\mathcal{E}\_g - E\_g| \le C\, \bigl(\tfrac{1}{\sqrt{N}} + \tfrac{1}{\sqrt{P}}\bigr)\,E\_g
> \quad\text{(with high probability)}, $$
> for constant $C$. We will clarify the notion of "deterministic" equivalence between the random quantity $\mathcal{E}\_g$ and the deterministic quantity $E\_g$ at large $P$ and $N$.
>
> >Eq 21 seems ... complete proof.
>
>  **Response:** The approximate equalities in Eq's C.15 - C.17 indicate that we are neglecting higher order terms in $\lambda$ and $1/N$. The result is a rigorous first-order approximation in these variables.  To clarify, we will replace $\approx$ with $\asymp$ (equal to leading order) in the derivation.
>
> > Appendix B.1... pointing out.
>
> **Response:** Thanks, we will clarify these steps in the SI.
>  For increasing $N$ we could argue as follows:
> Denote by $E_g(N, \lambda)$ the test with $N$ random features and ridge $\lambda$.  Consider $N \to N'$ for $N'>N$.  We show that there exists a ridge parameter $\lambda'$ such that $E_g(N', \lambda') \leq E_g(N, \lambda)$.  Next, we have that
> $$ \min_\lambda (E\_g(N', \lambda)) \leq E\_g(N', \lambda') \leq E\_g(N, \lambda)$$
> Because this is true for any $\lambda$, we may assign $\lambda = \operatorname{argmin}\_{\lambda} E\_g (N, \lambda)$, completing the proof.
>
> Analogous steps can be applied when increasing $P$ or for the joint transformation of $N$ and $K$ with $KN=M$.
>
> References:
>
> [1] Defilippis et. al https://arxiv. org/abs/2405.15699.
>
> [2] Adlam et. al. https://arxiv.org/abs/2011.03321.

---

> > ### Comment · Reviewer_Ef22 · 2025-04-05
> >
> > (reposting this as a Rebuttal Comment, as I didn't realize that authors can't see Official Comments. Sorry about that!)
> >
> > Thanks for all the replies! I have a few follow-up points listed below:
> >
> > 1. On section 6: these updates sound great. In addition to the computational complexity being lowered from $O(M^3)$ to $O(M^3 / K^2)$, I totally agree with the point that ensembling allows parallelization. This is a really interesting statistical-computation tradeoff.
> >
> > 2. On Eq (7): To clarify, I'm confused because Eq 7 seems to be a function of Z. In particular, a random $Z$ defines a given $f(x)$. And Eq (7) refers to a fixed $f(x)$, that is, a fixed $Z$. On the other hand, the right hand side of Eq. (12) does not depend on $Z$, as it takes expectations over $Z$. I would believe that Eq. (7) will concentrate in $Z$ for large numbers of features. But then there needs to be some kind of limit taken here for Eq (7) and Eq (12) to both hold. But maybe I'm misunderstanding the point about concentration in $Z$.
> >
> > 3. On the high probability bound between $\mathcal{E_g}$ and $E_g$ -- are the commas here typos? Should the bound be $C \left( \frac{1}{\sqrt{N}} + \frac{1}{P} \right) E_g$? Just want to make sure I'm following here! Also is there an exact reference from Defilippis et al. that has this result?
> >
> > 4. On the approximate equalities: is it correct to say that every approximate equality in the paper actually means "this is an equality when dropping terms that are of size $\lambda^2$ or $1/N^2$ or smaller?" If so, doesn't this assume we're working under an asymptotic model that has $\lambda \to 0$? Why should we expect this?
> >
> > 5. One final thing from the review that wasn't addressed: what does it mean that $1/N$ and $\lambda$ are "[assumed] to be on the same order of magnitude"? This sounds like the imagined asymptotic model not only has $\lambda \to 0$, but also has $N \to \infty$ *and* they're going at about the same rate. Can the authors give some more clarification here?
> >
> > Thank you!

---

> > > ### Author Response · Authors · 2025-04-07
> > >
> > > Thank you for your constructive feedback, which has helped us improve the clarity of our results.  We respond to your comments below:
> > >
> > > > On section 6: ... statistical-computation tradeoff.
> > >
> > > **Response**: We are glad that you agree that this is an interesting question, and will emphasize these points about parallelization and improved computational complexity in ensembles in the final version if accepted.
> > >
> > > > On Eq (7): To clarify, I'm confused because Eq 7 seems to be a function of Z...
> > >
> > > **Response**: Thank you for your comment, which we did not have space to fully address in our last reply. We will make some edits to section 2 to clarify the meaning of the errror formulas, and to make sure that the meaning of $E_g^K$ is consistent throughout.
> > >
> > > First, we will replace the $E_g^1$ in equation 7, with the "true" error symbol $\mathcal{E}_g^1$ to indicate that this depends on a particular realization of $\mathbf{Z}$:
> > >
> > > $$ \mathcal{E}\_g^1=\mathbb{E}\_{\boldsymbol{x} \sim \mu\_{\boldsymbol{x}}}\left[\left(f(\boldsymbol{x})-f\_*(\boldsymbol{x})\right)^2\right]+\sigma\_\epsilon^2 \qquad (7)$$
> > >
> > > We will then introduce the *deterministic equivalent* error $E_g^1$, which is the deterministic quantity about which the "true" error $\mathcal{E}_g^1$ concentrates.  This satisfies
> > >
> > > $$ |\mathcal{E}\_g^1 - E\_g^1| \le C \bigl(\tfrac{1}{\sqrt{N}} + \tfrac{1}{\sqrt{P}}\bigr) E\_g^1
> > > \quad\text{(with high probability)}, $$
> > >
> > > (commas in last reply were a formatting error).  As a shorthand, we may write $\mathcal{E}_g \simeq E_g$, with $\simeq$ indicating deterministic equivalence with a multiplicative error bound of this type.  We will update equation 9 so that it reads as:
> > >
> > > $$ \mathcal{E}\_g^1 \simeq E\_g^1 = \frac{1}{1-\gamma\_1}\left[-\rho \kappa_2^2 \mathrm{tf}\_1^{\prime}\left(\kappa\_2\right)+(1-\rho) \kappa\_2 \mathrm{tf}\_1\left(\kappa\_2\right)+\sigma\_\epsilon^2\right]  \qquad (9)$$
> > >
> > > So, the error formula on the right of eq. 9 is exactly equal to $E_g^1$.  $E_g^1$ is, in turn, a deterministic equivalent approximation of the "true" error $\mathcal{E}_g^1$.  We will similarly have $\mathcal{E}_g^K \simeq E_g^K$ for $K>1$.
> > >
> > > >On ... from Defilippis et al. that has this result?
> > >
> > > **Response**: Defilippis et. al. indeed is the reference with this exact result.  See equation 3 in their paper.
> > >
> > > >On the approximate equalities: ... we expect this?
> > >
> > > **Response**: No, it would not be correct to say that every approximate equality in our paper corresponds to dropping higher order terms in $\lambda$ and $1/N$.  There are two different types of approximation we are making.  The first is to replace the random quantity $\mathcal{E}_g^K$ with it's deterministic equivalent $E_g^K$.  All of our results rest on this simplification, which is justified due to the multiplicative concentration bounds proven by Defilippis et. al, and by our numerical simulations.  Theorems 4.1, 5.1, and corollary 5.5 apply exactly to $E_g^K$ with no further approximations, and with no assumptions about the ridge.
> > >
> > > Additional approximations are used in In sections 5.1 and 6 to study the behavior of $E_g^K$ in limits corresponding to the overparameterized and underparameterized regimes, respectively.
> > >
> > > In section 5.1, we expand $E_g^K$ in the limit $N \gg 1$, keeping $P \sim \mathcal{O}(1)$, so that $N \gg P$ (overparameterized).  Equation 21 is the **only** equation in the paper that assumes an asymptotically small ridge parameter $\lambda \approx 0$, and this is indicated in the correction term $\mathcal{O}(\lambda^2, \lambda/N, 1/N^2)$.  Here, we have assumed small ridge for analytical convenience, but the result (eq. 21) provides a good explanation of our empirical results in fig, S2.C and S2.F at optimal ridge as well.  Specifically, the overlap between the green curves ($P = 10$) and dashed black lines show that ensembles achieve *near-optimal* performance in the heavily overparameterized regime.  Simon et. al. (2023) also provide an argument for why optimal ridge is always small in the overparameterized regime (see theorem 2 there).  We include discussion of this in the final version of the paper.
> > >
> > >
> > > > One final thing ...  "[assumed] to be on the same order of magnitude"? ... more clarification here?
> > >
> > > **Response**: To clarify this point, we will remove the statement that we are assuming that $\lambda$ and $1/N$ are "on the same order of magnitude," and instead use the terminology "we expand as power series in $1/N \gtrsim 0$ and $\lambda  \approx 0$."  Again, this only applies to the derivation of eq. 21, and higher-order contributions are accounted for in the additive correction term there ($\dots + \mathcal{O}(\lambda^2, \lambda/N, 1/N^2)$).
> > >
> > > > To conclude
> > >
> > > Since this is our last opportunity to reply, we want to again thank the reviewer for giving such a careful reading of our paper, and for helping us improve the clarity of our presentation.

---

### Official Review · Reviewer_6CpK · 2025-03-13

**Overall Recommendation:** 4

**Summary:**

In the context of random feature high-dimensional ridge regression, this paper investigates the problem of training an ensemble of independent models and the trade-off between ensemble size and model size for a fixed total number of features. The authors prove a 'no free lunch' theorem, showing that increasing the ensemble size while keeping the total number of features fixed leads to higher test risk, making a single large model the optimal choice, provided the ridge parameter is fine-tuned. However, in the overparameterized regime, small ensembles can achieve near-optimal performance over a wider range of ridge parameter values, making them more robust than single models. The authors derive scaling laws showing that while optimal error scaling is always achieved by increasing model size with a fixed ensemble size, near-optimal scaling can be achieved under certain conditions on the kernel and task eigenstructure. These findings are validated through numerical simulations on synthetic data and real-world datasets.

**Claims And Evidence:**

All claims are supported by clear and convincing evidence.

**Essential References Not Discussed:**

I am not aware of any relevant references that have been omitted.

**Experimental Designs Or Analyses:**

I have no issues to discuss.

**Methods And Evaluation Criteria:**

The proposed methods are well-suited for the problem considered.

**Other Comments Or Suggestions:**

Typo on line 121, second column: "et al." is repeated twice

**Other Strengths And Weaknesses:**

This work improves the current understanding of generalization and scaling laws for RFRR, extending previous findings by tackling the problem of ensemble learning, which is closely related to practical issues and applications. The presentation is clear and provides valuable insights that could apply to other settings, such as explaining the apparent violation of the "no free lunch from ensembles" principle. Moreover, all claims are supported by extensive numerical simulations. I have not identified any major weaknesses.

**Questions For Authors:**

I have no important questions.

**Relation To Broader Scientific Literature:**

This paper builds on prior works showing that single large models can outperform ensembles when optimally tuned and extends the theoretical understanding of random feature generalization to the ensemble setting. The authors also contribute to the literature on scaling laws by deriving how test risk scales with ensemble and model size, providing insights into the trade-offs between the two quantities.

**Theoretical Claims:**

I checked the correctness of the proofs and I have no issues to discuss.

---

> ### Author Rebuttal · Authors · 2025-04-01
>
> Thank you for your review!

---

### Official Review · Reviewer_2SLL · 2025-03-14

**Overall Recommendation:** 2

**Summary:**

The paper investigates the performance of random feature ensembles, and discusss whether the ensemble models outperform the single model when the number of total parameter is fixed. The theoretical analysis is given based on the random feature ridge regression, while the empirical studies are performed on binarized CIFAR10 task. The paper provides two main results, the first one shows the RF ensemble models always benefit from the larger ensemble member and the more ensembles, while the second shows when there is a fixed total parameter count, increasing the number of ensemble size $K$ degrades the performance with an optimal ridge parameter. Moreover, the paper also derives a acaling law for the underparameterized ensembles.

**Claims And Evidence:**

The paper provides the sufficient theoretical analysis to support the claims, and the experiments based on the random feature ridge regression on binarized CIFAR10 task also shows the convincing results.

**Essential References Not Discussed:**

No additional references need to be discussed here. The authors have covered all the essential studies in the experiments or the related work.

**Experimental Designs Or Analyses:**

The experimental studies are sufficient and verify the claims in this paper. The Figure 1 shows increasing the number of samples P and the ensemble member size N can reduce the predictor error, while the Figure 2 shows increasing the number of ensemble menbers K while fix the total number of random features M, the performance of ensemble model degrades.

**Methods And Evaluation Criteria:**

The theorems and results provided in this paper are easy to understand, and there are several works referenced in this paper already give the similar analysis that decreasing the random features $N$ leads to a poor performance. The proposed results are clear and make sense for the emsemble RF models.

**Other Comments Or Suggestions:**

This is a paper with limited contributions, the author should provide more interesting findings, such as extending the results on deep learning area or other ensembel models.

**Other Strengths And Weaknesses:**

Weaknesses:
1. The contribution of this paper is limited, the results in this paper are evident, and can be easily derived based on the prior works.
2. The results mainly based on the random feature ridge regression model, whether the results suit for the deep learning models are not discussed.
3. The layout of the figures in this paper is irregular, as well as the fonts in these figures.

**Questions For Authors:**

Most of the experiments are based on the random feature RF model. Whether the same results shows in some larger ensemble models?

**Relation To Broader Scientific Literature:**

The results presented in this paper is clear and evident, the theoretical analysis is mainly based on the prior works, and the results are mainly focus on the random feature ridge regression model, whether the results are useful for the other machine learning models or deep neural networks are unclear.

**Theoretical Claims:**

The paper presents the theoretical analysis to prove the theorems, and the claims is proved based on the prior works and some obvious inequations.

---

> ### Author Rebuttal · Authors · 2025-04-01
>
> # Rebuttal to Reviewer 2SLL (Complete)
>
> Thank you for your review and questions.  It appears that you are convinced of the correctness of our results, but have concerns about the significance of the contribution.  We will respond to your concerns and questions individually below:
>
> >1. The contribution of this paper is limited; the results are somewhat evident and can be derived from prior works.
>
> **Response:**  The error formulas for RFRR ensembles derived in previous works are not easily interpretable, and studying the implications of these deterministic equivalent errors is an important task unto itself.  We believe that our paper has addressed an important gap in our understanding of ensembled random feature models. We have used the (known) bias-variance decomposition of the test risk of RFRR to study the tradeoff between model size and ensemble size.  Specifically, we have shown that:
> - Ensembling is *never* optimal under a fixed parameter budget at optimal ridge.
> - Ensembling can achieve near-optimal performance in both the overparameterized and underparameterized regimes, with precise spectral conditions for near-optimal scaling in the underparameterized regime.
> If accepted, we will clarify these contributions in the abstract and introduction.  We may also change the title of the paper to "No Free Lunch from Random Feature Ensembles: Scaling Laws and Near-Optimality Conditions" to highlight our contributions beyond Theorem 4.1.
>
> >2. The results mainly pertain to the random feature ridge regression model; whether they apply to deep learning models remains unaddressed.
>
> **Response:** While we are ultimately interested in understanding the utility of ensemble learning using state-of-the-art machine learning models, random-features regression provides an important "ground-floor" to investigate the utility of ensemble learning in a tractable setting.  Furthermore, because of the approximate correspondence between RFRR and deep networks trained in the lazy learning regime [3], our results already bear some relevance to deep ensembles.  Understanding the properties deep ensembles trained in the rich regime is a critical research direction for our future work, but will be better presented with reference to these baseline results in linear models, which have already saturated the page limit for this venue.
>
> Our results also add to a long history of research on random-feature models in analogy to deep networks. Another example of this fruitful correspondence is work on fine-grained bias-variance decompositions [2]
>
> We have also performed numerical experiments for deep ensembles.  In these experiments, we train ensembles of deep convolutional neural networks on a computer vision task (CIFAR10 image classification) using $\mu P$ parameterization, which keeps training dynamics consistent across widths [1].  In addition to the weight decay, there is a "richness" parameter $\gamma$ which controls the amount of feature-learning in the network.  Our simulations show that a large network outperforms any ensemble of smaller networks with the same total size when both the weight decay and "richness" parameter are tuned to their optimal values.  If it will have a positive impact on your evaluation, we are willing to add these numerical results to the supplemental material.  This result is available at this anonymized github repo: https://anonymous.4open.science/r/NoFreeLunchRandomFeatureEnsembles/README.md
>
> >3. The layout of the figures is irregular, and the fonts in these figures could be improved.
>
> **Response:** Thank you for your feedback, we are open to any and all suggestions on how to improve the clarity and appearance of our figures for the final version of this paper if accepted!
>
> > "Most of the experiments are based on the random feature (RF) model. Would the same results hold for larger ensemble models?"
>
> **Response:** See our response to 2. above.
>
>
> [1] Nikhil Vyas, Alexander Atanasov, Blake Bordelon, Depen Morwani, Sabarish Sainathan, and Cengiz Pehlevan. Feature-learning networks are consistent across widths at realistic scales, 2023. URL https://arxiv.org/abs/2305.18411.
>
> [2] Ben Adlam and Jeffrey Pennington. Understanding double descent requires a fine-grained bias-variance decomposition, 2020. URL https://arxiv.org/abs/2011.03321.
>
> [3] Chizat, L., Oyallon, E., & Bach, F. (2020). On lazy training in differentiable programming (arXiv:1812.07956v5). Retrieved from https://arxiv.org/abs/1812.07956

---

### Decision · Program_Chairs · 2025-05-01

**Decision:**

Accept (poster)

**Comment:**

This work provides an asymptotic comparison of random feature ensembles versus a single large random feature regressor under a fixed feature budget. While the authors demonstrate that no ensemble will outperform a single regressor with an optimally tuned ridge parameter, they demonstrate conditions under which an ensemble is near optimal. This work is theoretically sounds and builds upon many recent derivations of the asymptotic risk of random feature regression. While some portions of the theoretical argument were not rigorously presented in the original submission, the authors have addressed these concerns during the discussion period. I urge the authors to incorporate these suggestions for clarity and rigour in the camera ready. Overall, this work tackles a relevant problem and adds to the knowledge of scaling laws, and therefore I recommend acceptance.